# Enterocytes rely on purine biosynthesis/salvage pathway to facilitate dietary fat absorption

Yu Wang [1] ✉, Li Chen[2,3], Yingze Ma [1], Mingqi Zhou [4], Aleksander Geske[5], Marcus Seldin [4], Jiangjiang Zhu [2,3], Alexander M. Zak[1], Xinzhong Dong [5,6], Robert N. Cole [7] & Svetlana Lutsenko [1] ✉

Dietary fat absorption is among the most energy-demanding processes of nutrient uptake. Fatty acid activation, triglyceride synthesis, and the trafficking of chylomicrons through the secretory pathway - all require ATP. How enterocytes accommodate the surge in ATP consumption following fat uptake is unclear. We show that the purine biosynthesis/salvage pathway supplies necessary ATP and that Ankyrin Repeat Domain 9 (ANKRD9) couples ATP synthesis and lipoprotein trafficking. Ankrd9 regulates enzymes within the purine biosynthesis pathway to increase ATP synthesis and facilitate Golgi dynamics. Intracellular localization of ANKRD9 is lipid and ATP-dependent. Inactivation of Ankrd9 in mice reduces intestinal ATP despite intact mitochondrial and glycolytic function, alters Golgi morphology, delays ApoB/chylomicron trafficking, and causes lipid accumulation in enterocytes, along with a lean body phenotype. Taken together, the results reveal a previously unrecognized mechanism that regulates lipid absorption in enterocytes and identify ANKRD9 as a central component of this mechanism.

Energy production and utilization are fundamental properties of every cell. Cellular processes that generate ATP, such as glycolysis, mitochondrial respiration, and the de novo purine synthesis/salvage, are well characterized[1–3]. Changes in nucleotide levels provide regulatory feedback to many biosynthetic enzymes to maintain nucleotide balance[4,5]. The consumption of ATP markedly increases during dietary fat absorption to fuel the production and trafficking of chylomicrons, the major carriers of dietary fat. The mechanism that accommodates a rapid increase in energy expenditure by the gut during dietary fat absorption is unknown. By investigating the role of the Ankyrin Repeat Domain 9 (ANKRD9) protein in the small intestine, we discovered a critical role of ANKRD9 in maintaining high ATP levels, necessary for the efficient intracellular processing of chylomicrons. We dissected the

mechanism of ANKRD9 activity using genetically engineered mice, intestinal organoids, and cultured human cells, which we examined using fluorescence microscopy, metabolite analysis, proteomics, and in silico assays.

ANKRD9 is a highly conserved protein with approximately 85% sequence identity between humans and mice. It belongs to a large family of proteins containing multiple ankyrin repeats folded into a solenoid-like structure, known as an ankyrin repeat domain (Fig. 1a). These domains are typically engaged in protein-protein interactions[6–8]. High conservation of ANKRD9 among different species and the lack of sequence similarity to other proteins suggest a unique and important function of this protein. However, current information about ANKRD9 is limited and confusing. It was shown that ANKRD9 mRNA is

[1]Department of Physiology, Pharmacology & Therapeutics, Johns Hopkins University School of Medicine, Baltimore, MD, USA. [2]Human Nutrition Program, Department of Human Sciences, The Ohio State University, Columbus, OH, USA. [3]James Comprehensive Cancer Center, The Ohio State University, Columbus, OH, USA. [4]Department of Biological Chemistry, University of California School of Medicine, Irvine, CA, USA. [5]Solomon H. Snyder Department of Neuroscience, Johns Hopkins University School of Medicine, Baltimore, MD, USA. [6]Howard Hughes Medical Institute, Chevy Chase, MD, USA. [7]Department of Biological Chemistry, Johns Hopkins University School of Medicine, Baltimore, MD, USA. ✉e-mail: ywang391@jh.edu; lutsenko@jhmi.edu

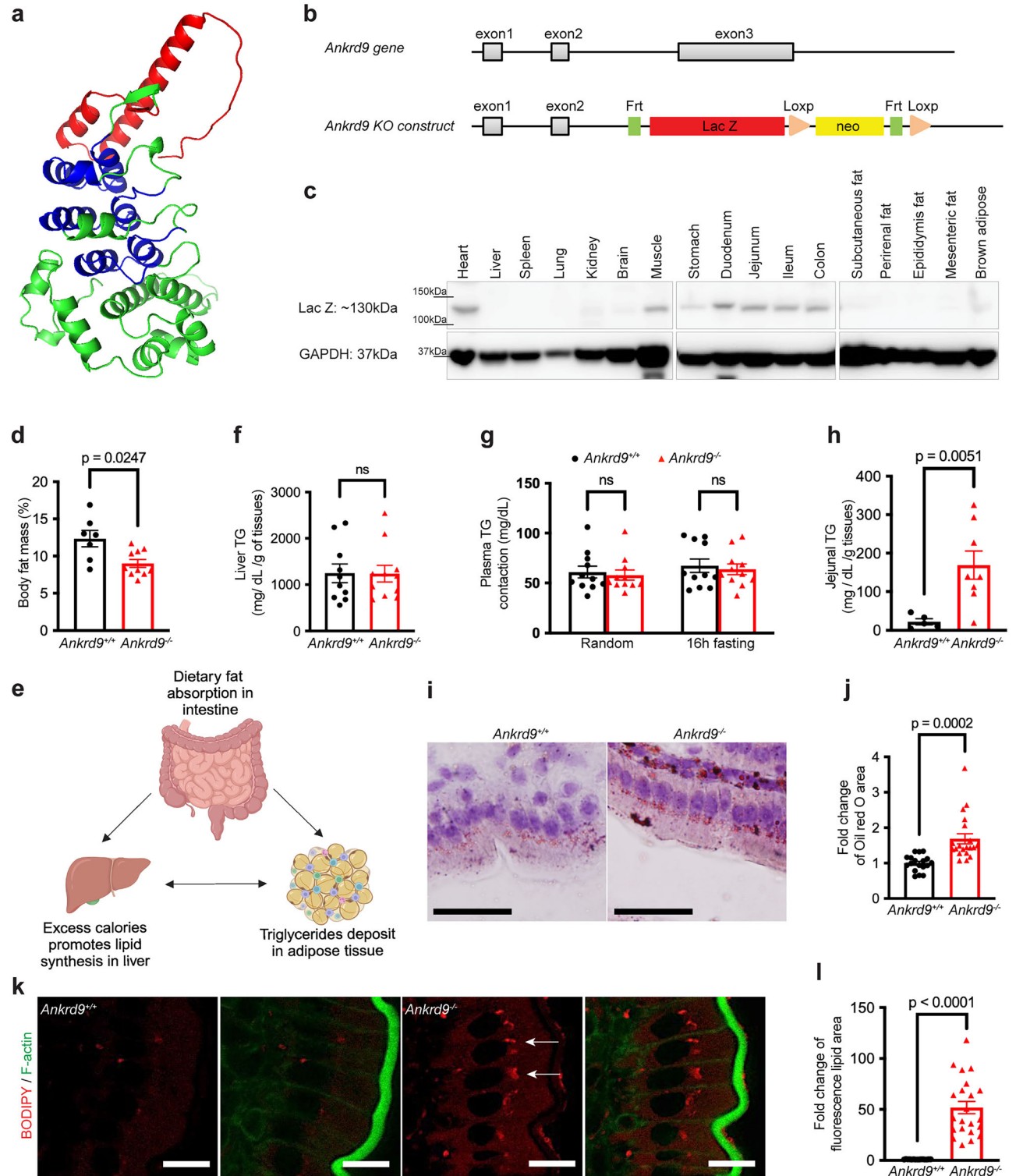

**Fig. 1 | Ankrd9 is enriched in metabolically active tissues and is involved in intestinal lipid metabolism. a** AlphaFold model of human ANKRD9 (Q96BM1). Ankyrin repeat domain is in blue; the 63 N-terminal residues are in red; the rest is in green. **b** Schematic of the *Ankrd9* gene (top) and the construct used for generating the Ankrd9 knockout mice (bottom); neo: neomycin resistance cassette. **c** Western blot analysis of LacZ expression (Ankrd9) in different tissues of *Ankrd9*⁻/⁻ mice; Uncropped images of blots are shown in the Source Data File; Data represent three independent experiments. **d** Total fat mass in 14-week-old *Ankrd9*⁺/⁺ and *Ankrd9*⁻/⁻ mice determined by magnetic resonance imaging; *n* = 7 and 10 mice in *Ankrd9*⁺/⁺ and *Ankrd9*⁻/⁻ groups, respectively. **e** Schematic of major sources of triglycerides in the body; Created in BioRender. Wang, Y.(2025) https://BioRender.com/c18ukeh.

Triglyceride levels in the liver (**f**), plasma (**g**) and jejunum (**h**) in *Ankrd9*⁺/⁺ and *Ankrd9*⁻/⁻ mice; *n* = 10 - 11 mice per group in (**f**), *n* = 11 mice per group in (**g**), *n* = 5–8 mice per group in (**h**), please refer to the Source Data File. **i** Jejunum stained with Oil red O (red); Scale bar: 100 μm; *n* = 4 *Ankrd9*⁺/⁺ and 5 *Ankrd9*⁻/⁻ mice. **j** Quantitation of Oil red O area fold change in (**i**), *n* = 18 sections from 4 *Ankrd9*⁺/⁺ and 20 sections from 5 *Ankrd9*⁻/⁻ mice. **k** BODIPY staining (red) of jejunum; Scale bar: 10 μm; *n* = 5 *Ankrd9*⁺/⁺ and 6 *Ankrd9*⁻/⁻ mice (the white arrow indicates accumulated lipids), and (**l**) Quantitation of BODIPY fluorescence area per cell, fold change compared to *Ankrd9*⁺/⁺ cells, *n* = 15 sections from 5 *Ankrd9*⁺/⁺ and 22 sections from 6 *Ankrd9*⁻/⁻ mice. The tissues were collected under non-fasting conditions. (**d**, **f**–**h**, **j**, **l**) *p*-values as indicated by a two-tailed unpaired t-test; Data are presented as the mean ± SEM.

responsive to changes in fat metabolism[9], that ANKRD9 can act as an adaptor for ubiquitin ligase[10,11], regulates inosine-5′-monophosphate dehydrogenase 2 (IMPDH2) stability[10,12], and contributes to cellular copper homeostasis[13]. The physiological relevance of these disparate findings has not been explored in either primary cells or in vivo; and the role of ANKRD9 (Ankrd9 in mice) in mammalian physiology remains unclear.

Here, we have identified the central role of ANKRD9 in a regulatory mechanism that facilitates dietary fat absorption in the small intestine. We demonstrate that ANKRD9 is enriched in tissues with high energy demands and, in the intestine, it co-regulates purine biosynthesis/salvage pathway and lipoprotein trafficking. Specifically, ANKRD9 shifts purine flux toward ATP synthesis, facilitates Golgi network dynamics, and accelerates lipoprotein trafficking. Deletion of Ankrd9 reduces intestinal ATP despite intact mitochondrial and glycolytic function, delays ApoB/chylomicron trafficking, and causes lipid accumulation in enterocytes and a lean body phenotype. These findings identify purine metabolism as a dedicated energy source for efficient fat processing in enterocytes, and establish ANKRD9 as a molecule coordinating ATP availability with chylomicron trafficking through the secretory pathway.

## Results

### ANKRD9 is expressed in metabolically active tissues

To understand the physiologic role of ANKRD9, we used Ankrd9 knockout mice (Ankrd9$^{-/-}$). In these mice, the largest of three Ankrd9 exons, exon 3, is replaced with Frt-LacZ-loxp-Neo-Frt-loxp cassette (Fig. 1b), resulting in the loss of Ankrd9 and the expression of LacZ under the endogenous Ankrd9 promoter. In the absence of reliable anti-mouse Ankrd9 antibodies, we used LacZ expression as a proxy for expression of Ankrd9 to identify tissues that require Ankrd9 function. LacZ signal was most abundant in the intestine, heart, and skeletal muscle (Fig. 1c). Similarly, Human Protein Atlas shows enrichment of ANKRD9 mRNA in human heart, skeletal muscle, and intestinal tissues (Supplementary Fig. 1). The common property of these tissues is their high metabolic activity and the ability to accommodate fluctuating energy demands[14–17]. Enrichment of Ankrd9 in these tissues suggested that Ankrd9 could be involved in regulating energy supplies and/or utilization. For subsequent experimental work, we focused on the small intestine, because the small intestine (jejunum) is central for the processing of dietary fat[18–20], and ANKRD9 was previously found upregulated when fatty acid metabolism was disturbed[9].

### Ankrd9 is required for the efficient processing of dietary fat

Adult Ankrd9$^{-/-}$ mice appear healthy and have the same body weight and random blood glucose level as control Ankrd9$^{+/+}$ animals; fasting blood glucose levels are lower than in controls (Supplementary Fig. 2a-d). Their insulin sensitivity is unaltered (Supplementary Fig. 2e). However, the total body fat mass in Ankrd9$^{-/-}$ male mice is significantly lower than in control mice (Fig. 1d). Two major sources of body fat are chylomicrons, which are produced by the small intestine following dietary fat uptake, and triglycerides (TG), which are synthesized and then released into the blood by the liver[21](Fig. 1e). In Ankrd9$^{-/-}$ mice, the TG levels in the liver and the blood are similar to those of control mice (Fig. 1f-g), whereas the TG levels in Ankrd9$^{-/-}$ intestine (jejunum) are 8-fold higher than in Ankrd9$^{+/+}$ controls (Fig. 1h). Staining with Oil Red O and BODIPY confirmed the higher TG content in Ankrd9$^{-/-}$ jejunum and detected lipid accumulation at the perinuclear region of enterocytes (Fig. 1i-l). Thus, dietary fats enter the jejunal enterocytes of Ankrd9$^{-/-}$ mice, but their further processing is altered.

### Fatty acid uptake is unaffected by Ankrd9 deletion

The intestine length and morphology in Ankrd9$^{-/-}$ mice are normal (Supplementary Fig. 3a-c); thus, the defects in Ankrd9$^{-/-}$ enterocyte function are likely intracellular. To identify which step of lipid processing was altered in Ankrd9$^{-/-}$ enterocytes, we examined lipid uptake, the abundance of ApoB, which is the main protein component of chylomicrons[22–24], and the trafficking of ApoB/chylomicrons through the secretory pathway. The ApoB and lipid trafficking in mouse enterocytes were analyzed in vivo, after 6 hours of fasting followed by the gavage with olive oil (Fig. 2a and Supplementary Fig. 4a). The BODIPY staining of the jejunum detected lipid-containing particles as early as 15 min after oil gavage in both control mice (Fig. 2b, two top rows) and Ankrd9$^{-/-}$ mice (Fig. 2b, two bottom rows). The lipid staining was stronger in Ankrd9$^{-/-}$ enterocytes, which may reflect incomplete removal of pre-existing TG pools by a 6-hour fasting (Fig. 2b-c). At 30 min, lipid signal was also stronger in Ankrd9$^{-/-}$ mice, however, within 60 min, the amount of fat that entered the control and mutant intestine was comparable (Fig. 2b-c).

To directly compare the rates of lipid uptake, we measured time-dependent changes in the amount of oleic acid (C18:1 fatty acid (FA), the major constituent of olive oil) in the jejunum using mass spectrometry. The rate of C18:1 FA accumulation was similar for the control and Ankrd9$^{-/-}$ intestine (Fig. 2d), confirming that a stronger BODIPY staining in Ankrd9$^{-/-}$ mice was due to a preexisting lipid pool. We found no difference in the rate of appearance of (C18:1)-based diacylglycerides (DG) (Supplementary Fig. 4b). The TG content in the control jejunum increased steadily over time in parallel with fatty acid uptake. The Ankrd9$^{-/-}$ jejunum showed an apparent decrease in TG at 15 min after oil gavage (statistically not significant, p-value = 0.096), whereas at later time points the TG (C18:1) reached levels similar to wild-type (Supplementary Fig. 4c). Taken together, the results suggest that the loss of Ankrd9 does not compromise the jejunum's ability to uptake fatty acids or synthesize TG but the initiation of TG synthesis may be delayed.

### Trafficking of ApoB is altered in Ankrd9$^{-/-}$ intestine

The protein and mRNA levels of ApoB were unaffected by Ankrd9 inactivation (Supplementary Fig. 4d-f), whereas the localization and trafficking of ApoB were changed significantly (Fig. 3a). In the wild-type jejunum, ApoB was mostly concentrated at the perinuclear region of enterocytes (Fig. 3b). This targeting is likely to reflect the known localization of ApoB in rough endoplasmic reticulum (ER), where it accepts incoming lipids to form pre-chylomicrons, and in the trans-Golgi network (TGN), where pre-chylomicrons mature before trafficking to the lateral membrane[22,25–27]. Within 15 min of lipid ingestion, ApoB moves out of the perinuclear compartment and disperses through the cytosol, as evident from the loss of perinuclear signal (Fig. 3c-d). At 30 min, ApoB concentrates near the apical membrane, in agreement with the previously reported data[28]. At 60 min, ApoB's abundance in the vicinity of the apical membrane decreases and ApoB appears at the lateral membrane - consistent with the export of mature chylomicrons. At 90 min, ApoB returns to its original distribution pattern (Fig. 3c-d).

In Ankrd9$^{-/-}$ jejunum before fat uptake, ApoB is found mostly in vesicles distributed throughout the cell, and the perinuclear signal of ApoB is minor. At 15 min after the oil gavage, ApoB concentrates in the perinuclear compartment and, at 30 min, it moves out of this compartment (Fig. 3c-d). ApoB accumulation at the apical membrane is delayed to 60 min, and the staining of ApoB there is less intense (Fig. 3c-d). At 90 min, ApoB signal is seen at the lateral membrane, although with lower intensity compared to controls, whereas the strong ApoB signal appears near the lipid-containing particles (Fig. 3c-d). Thus, the overall ability of ApoB to traffic through the secretory pathway is not interrupted by the loss of Ankrd9. Rather, the delivery of ApoB to the apical membrane and the lateral membrane is delayed, resulting in a predominantly vesicular pattern of ApoB at a steady state and a slower trafficking in response to lipids.

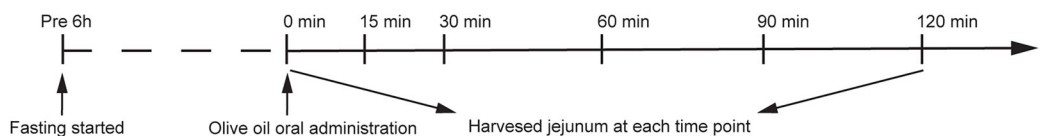

**Fig. 2 | Ankrd9 deletion does not decrease lipid uptake and the ApoB abundance. a** Schematic of lipid treatment experiment using olive oil gavage. **b** Lipid processing in *Ankrd9+/+* and *Ankrd9−/−* jejunum during oil treatment as revealed by BODIPY fluorescence (green); Scale bar: 10 µm, *n* = 3 mice per each time point per group per experiment; Data represent three independent experiments. **c** Quantitation of BODIPY fluorescence (per cell), fold change compared to time zero; *n* = 5–19 sections per time point per group, please refer to the Source Data File. **d** Mass-spectrometry measurements of oleic acid (C18:1-FA) levels in jejunum during olive oil treatment; values were normalized to tissue weight; *n* = 3–5 mice per time point per group at each time point, please refer to the Source Data File. **c**, **d** *p*-values as indicated by two-way ANOVA with Šídák test; Data are presented as the mean ± SEM.

## Ankrd9 does not interact with ApoB but clusters near the cis-Golgi and the lateral membrane

Given ApoB mis-localization and trafficking delays in *Ankrd9−/−* mice, we tested whether Ankrd9 regulates ApoB through direct protein-protein interactions; to accommodate the antibodies' species specificity, these studies were done in human differentiated Caco-2 cells, which express ApoB (Supplementary Fig. 5a). We observed no colocalization between ANKRD9 and ApoB (Supplementary Fig. 5b), indicating that interactions are absent or transient. Previously, the recombinant ANKRD9 expressed in HEK293 cells was found in puncta, which were spread throughout the cytosol and did not overlap with the markers of major cell compartments[10,12]. However, in differentiated

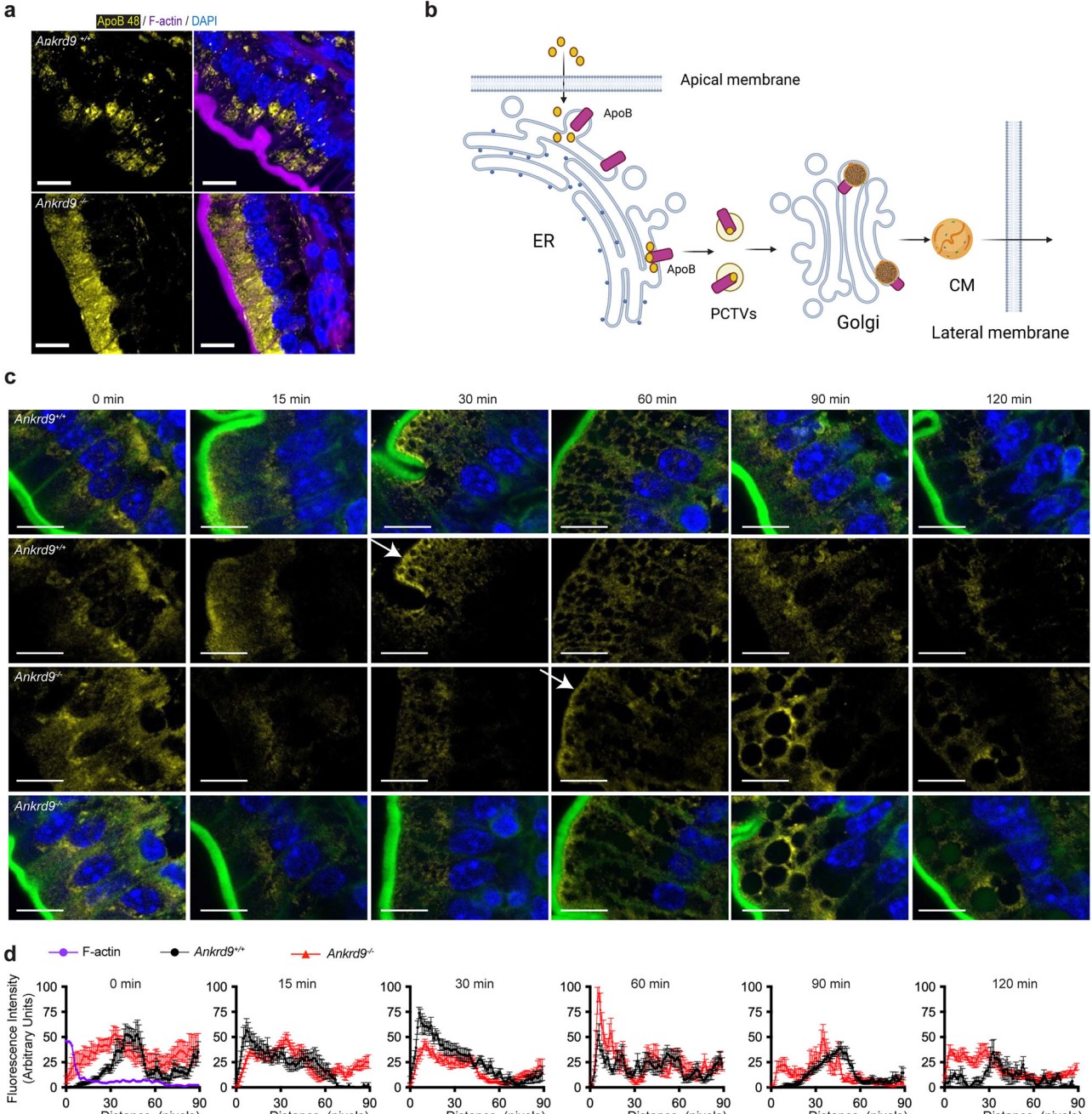

**Fig. 3 | Ankrd9 deletion alters ApoB trafficking in vivo. a** Immunostaining of ApoB (yellow) in *Ankrd9⁺/⁺* and *Ankrd9⁻/⁻* jejunum; the tissues were collected under non-fasting conditions; *n* = 10 mice per group. Scale bar: 10 μm. **b** Schematic of lipid (yellow circles) and ApoB (purple cylinders) trafficking in enterocytes; Created in BioRender. Wang, Y.(2025) https://BioRender.com/re6svli. ER: endoplasmic reticulum; PCTV: pre-chylomicron transport vesicles; CM: chylomicron; **c** Immunohistochemical staining of ApoB (yellow) in *Ankrd9⁺/⁺* and *Ankrd9⁻/⁻* jejunum during lipid processing; time points as in Fig. 2a; F-actin (green) is the apical membrane marker. *n* = 3 mice per each time point per group per experiments; The data represent three independent experiments. Scale bar: 10 μm; The white arrows indicate ApoB at the apical membrane. **d** Quantitation of cellular ApoB distribution in Fig. 3c, using the RGB profile plot from the ImageJ software package; Distance from F-actin, whose location is taken as a zero; *n* = 4–14 sections at each time point per group, please refer to the Source Data File; Data are presented as the mean ± SEM.

intestinal Caco-2 cells grown on Trans-wells[29], ANKRD9 puncta have a distinct pattern: they cluster in proximity to the cis-Golgi marker GM130 and near the apical membrane (Fig. 4a).

To better understand whether ANKRD9 is targeted to the apical membrane, we co-stained ANKRD9 with F-actin in differentiated and polarized enteroids (Fig. 4b). In enterocytes, F-actin forms a dense meshwork at the apical membrane and perijunctional actomyosin ring at the lateral membrane, which shows a characteristic "chicken-wire" pattern. The Z-sectioning and confocal microscopy of enteroid monolayers clearly demonstrate the absence of ANKRD9 at the apical membrane. Instead, we observed ANKRD9 subapical puncta and proximity of ANKRD9 to the F-actin-positive compartment of the lateral membrane (Fig. 4b and Supplementary Fig. 6a). Co staining with GM130 confirmed the location of intracellular ANKRD9 puncta near cis-Golgi (Supplementary Fig. 6a). Similar pattern was observed in human organoids (Supplementary Fig. 7a). The specificity of ANKRD9 staining was confirmed by ANKRD9 deletion in enteroids and Caco-2 cells (Fig. 4b and Supplementary Fig. 7b) and by

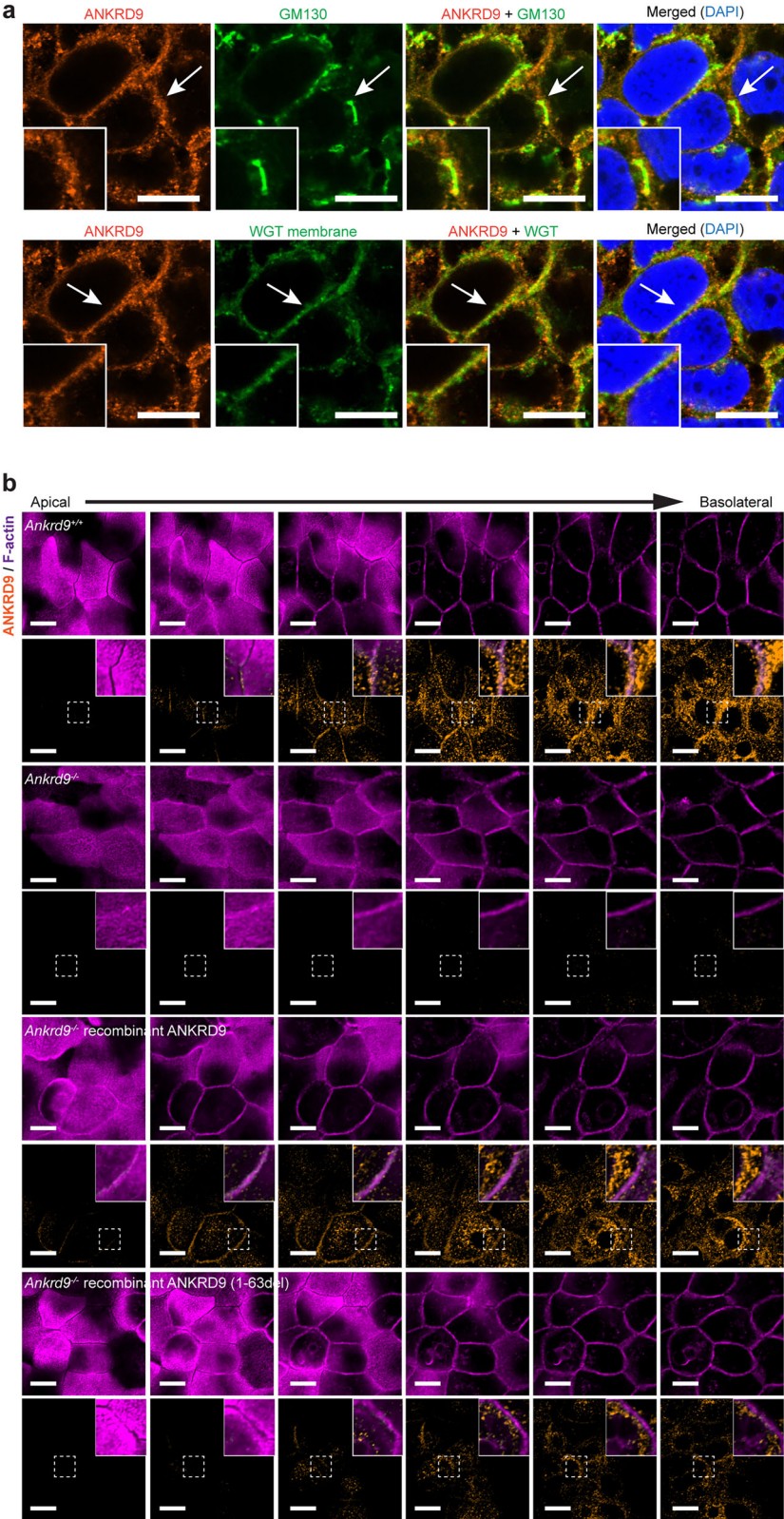

**Fig. 4 | The intracellular localization of Ankrd9. a** Co-immunostaining of ANKRD9 (orange) with the cis-Golgi marker GM130 (green) or with the plasma membrane marker wheat germ agglutinin (WGA, green) in Caco2 cells shows clustering of ANKRD9 puncta near the Golgi and the membranes. The white arrows indicate magnified regions at the lower left corner of each image.
**b** Immunostaining of ANKRD9 (orange) and F-actin (purple) in differentiated and polarized enteroids: Z-sectioning from the apical to basolateral aspect of the enteroid monolayer illustrates the absence of ANKRD9 at the apical membrane,

ANKRD9 proximity to F-actin at the lateral membrane, and intracellular puncta. The specificity of ANKRD9 patterns is confirmed by the loss of pattern in *Ankrd9*[−/−] enteroids and the restoration of the pattern following expression of recombinant ANKRD9 in *Ankrd9*[−/−] enteroids. The ANKRD9 (1-63del) variant shows intracellular puncta, but the membrane localization is decreased. The dotted boxes indicate the magnified regions at the upper right corner of each image. **a**, **b** Scale bar: 10 μm; Data represent three independent experiments.

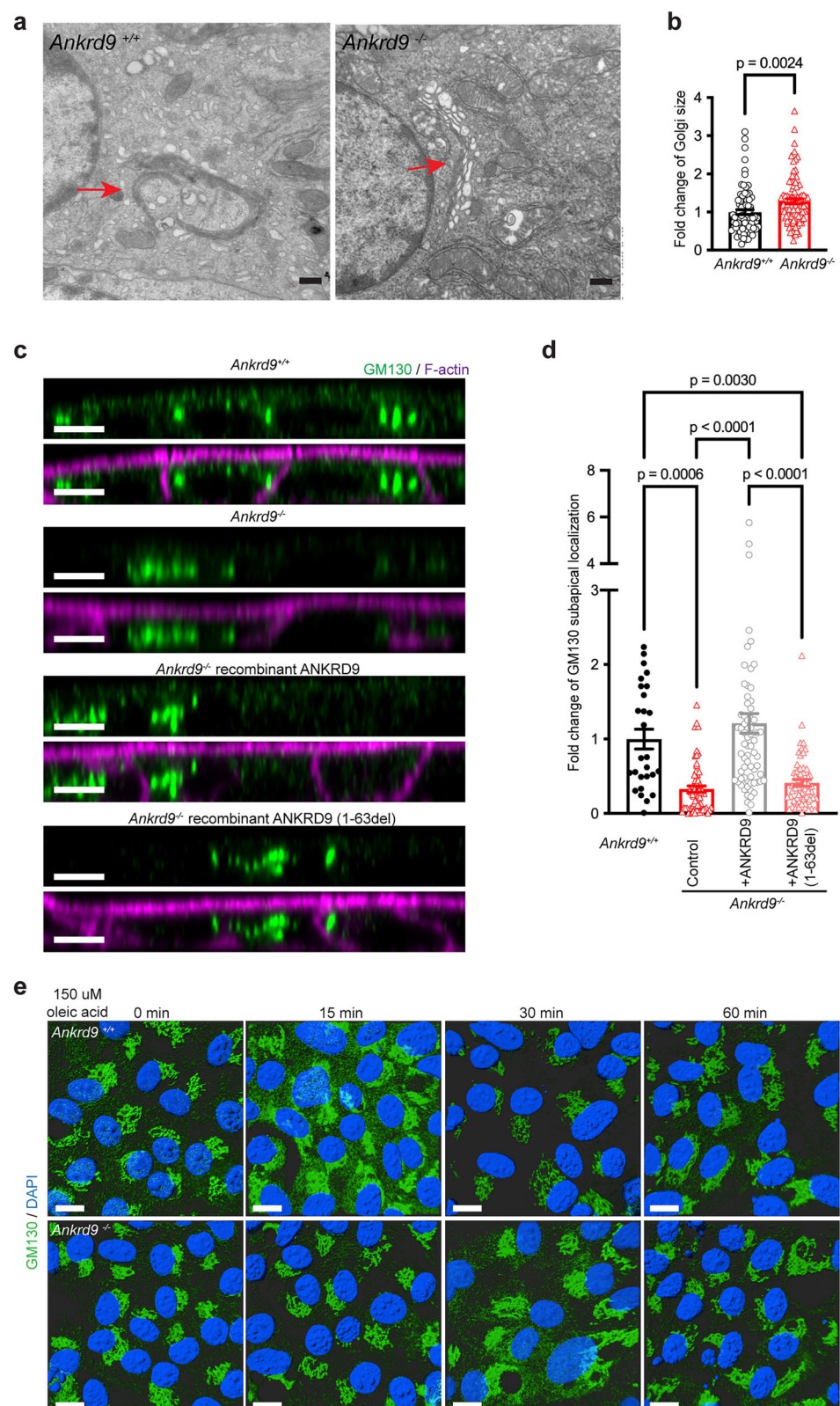

recombinant Flag-ANKRD9-mCherry expression (Fig. 4b and Supplementary Fig. 7c). Colocalization of ANKRD9 with the TGN marker, TGN46, was not detected (Supplementary Fig. 7d), highlighting the sidedness of ANKRD9 distribution with respect to the Golgi network.

By fractionating cell homogenates using differential centrifugation, we found that ANKRD9 forms large protein complexes, 440 kDa

and 160–250 kDa, found in the membrane-containing and cytosolic fractions in HEK293 and differentiated Caco2 cells, respectively (Supplementary Fig. 8a-d). Deletion of the N-terminal domain does not significantly change the size of these complexes, indicating that they are not ANKRD9 oligomers. However, the targeting to the lateral membrane is decreased by this deletion (Fig. 4b and Supplementary Fig. 6a-b).

**Fig. 5 | Deletion of Ankrd99 alters Golgi structure and dynamics. a** The EM images of the Golgi apparatus, each image represents an individual mouse; $n = 3$ mice per group; The red arrows indicate Golgi; Scale bar: 500 nm. **b** Size of the Golgi apparatus area relative to the average size of Golgi in control, $n = 82$ and 91 cells from *Ankrd9*[+/+] and *Ankrd9*[−/−] groups, respectively; p-values as indicated by a two-tailed unpaired t-test; Data are presented as the mean ± SEM. **c** Presence of subapical GM130-positive vesicles (green) correlates with expression of ANKRD9. The XZ images show numerous subapical GM130 signals in *Ankrd9*[+/+] enteroids along with the bright, centrally located Golgi staining. The subapical GM130 signal is decreased in *Ankrd9*[−/−] enteroids, but can be restored by the expression of

recombinant ANKRD9. Expression of the ANKRD9 (1-63del) variant in *Ankrd9*[−/−] enteroids does not restore the subapical GM130 signal; Scale bar: 5 µm. The data represent three independent experiments. **d** Qualification of the GM130-positive vesicles within 0.3 µm from the apical membrane; $n = 27$–71 cells per group, please refer to the Source Data File; p-values as indicated by two-way ANOVA with Tukey test; Data are presented as the mean ± SEM. **e** Transient expansion of cis-Golgi (marked by GM130, green) during treatment with oleic acid is delayed in *Ankrd9*[−/−] enteroids compared to control; Scale bar: 10um. The data represent three independent experiments.

## The morphology of the Golgi network is altered in Ankrd9[−/−] enterocytes

The location of ANKRD9 near the cis-Golgi suggested that it may facilitate the docking of ApoB-containing vesicles to the cis-Golgi and/ or contribute to the cis-Golgi to *trans*-Golgi trafficking of ApoB. To test this hypothesis, we first examined the status of intracellular compartments of *Ankrd9*[−/−] enterocytes using electron microscopy (EM) of the small intestine. There was no apparent difference in the abundance and morphology of the endoplasmic reticulum, indicating unperturbed ApoB and fat trafficking through this compartment. Mitochondria, nuclei, and plasma membrane were also unchanged (Supplementary Fig. 9a). In contrast, the size of the Golgi apparatus was significantly larger in *Ankrd9*[−/−] enterocytes compared to control cells. The cis-Golgi was especially enlarged, and extra vesicles were seen in the vicinity (Fig. 5a-b).

To further test the effect of ANKRD9 on the secretory pathway, we examined the Golgi morphology before and after oleic acid addition in control and *Ankrd9*[−/−] enteroid monolayer using confocal microscopy with Z-sectioning. At steady state, the enteroids grown on Trans-wells show immunostaining of GM130 in the perinuclear space, as expected, and also in a subset of vesicles near the apical membrane (Fig. 5c-d and Supplementary Fig. 6b). The *Ankrd9*[−/−] enteroid had abundant perinuclear GM130 staining, whereas the subapical vesicles were absent (Fig. 5c-d). Expression of the recombinant Flag-ANKRD9-mCherry in *Ankrd9*[−/−] enteroids restored the subapical GM130-positive pool of vesicles (Fig. 5c-d). In response to oleic acid, the Golgi network in control cells quickly becomes enlarged and disperse, but within 30 min the morphology is restored (Fig. 5e and Supplementary Fig. 9b). In *Ankrd9*[−/−] enteroid, this response is delayed and appears less pronounced (Fig. 5e). Taken together, the data suggest that the location of ANKRD9 near cis-Golgi contributes to the structural dynamics of this compartment.

## Deletion of Ankrd9 upregulates proteins involved in nucleotide synthesis and lipid transport

To gain insight into the mechanism of ANKRD9 action, we identified proteins and pathways affected by Ankrd9 deletion. Comparison of proteomes from *Ankrd9*[+/+] and *Ankrd9*[−/−] jejunal enteroids using a tandem mass tag (TMT)-labeling mass-spectrometry identified a total of 5693 proteins common for all biological and technical replicates. Among those, 15 proteins were altered in *Ankrd9*[−/−] enteroids more than 1.5-fold with a *p*-value < 0.05, when compared to control samples. These included proteins involved in purine homeostasis, glucose metabolism, lipid transport, protein trafficking (myosin 6, guanine exchange factors), and the components of ubiquitin ligase complexes (Fig. 6a-b and Supplementary Fig. 10a-b).

The most significant and consistent changes were observed for proteins maintaining nucleotide levels and lipid-transporting proteins. The abundance of IMPDH2 (the rate-limiting enzyme in GTP synthesis[30,31]) was increased more than 2-fold (Fig. 6b and Supplementary Fig. 10b) and this increase was verified by Western blotting of intestinal tissue homogenates (Supplementary Fig. 10c). In contrast, the abundance of adenylate kinase AK4 (involved in ATP homeostasis[32]) was decreased almost 2-fold (Fig. 6b). Lipid-carrying

proteins (fatty acid-binding proteins and the components of chylomicrons, ApoAI and ApoAIV) were significantly (1.6 ~ 1.9-fold) elevated in *Ankrd9*[−/−] samples (Fig. 6b), in agreement with intracellular lipid accumulation and the delay in chylomicrons maturation.

## Ankrd9[−/−] jejunum has low ATP despite normal mitochondrial respiration and glycolysis

Nucleotides (ATP and GTP) are essential for normal processing and trafficking of chylomicrons[33]. The major impact of Ankrd9 deletion on proteins associated with nucleotide metabolism and lipid transport suggested that Ankrd9 may co-regulate nucleotide homeostasis and chylomicron trafficking through the Golgi. To test this hypothesis, we measured the nucleotide levels in control and *Ankrd9*[−/−] mouse enteroid extracts using mass-spectrometry and found significantly lower ATP and GTP levels in *Ankrd9*[−/−] enteroids (decreased by 31% and 37%, respectively) when compared to controls (Fig. 6c, f). ADP and GDP were elevated (2-fold and 1.9-fold increase, respectively) (Fig. 6d, g), whereas AMP and GMP levels were unchanged by Ankrd9 deletion (Fig. 6e, h). Normally, ATP levels in cells exceed GTP levels by about 5-fold (Supplementary Fig. 10d), therefore depletion of cellular ATP likely represents the main physiologic effect of Ankrd9 inactivation, although lowering of GTP levels is also significant. We verified changes in ATP levels using the in vitro luciferase-based assay (Supplementary Fig. 11a).

The dramatic decrease in ATP levels was unexpected, and we examined the reasons for this decrease in more detail. Seahorse analysis found no changes in mitochondria respiration in *Ankrd9*[−/−] enteroids (Fig. 6i), in agreement with the unchanged abundance of most mitochondria proteins (Supplementary Fig. 11b). The decrease in AK4 abundance in *Ankrd9*[−/−] enteroids is likely a compensatory response to lower cytosolic ATP (Supplementary Fig. 11c), because down-regulation of AK4 can increase cellular ATP levels by up to 25%[32]. The abundance of rate-limiting enzymes involved in glycolysis, Krebs cycle and fatty acid oxidation was similar in *Ankrd9*[+/+] and *Ankrd9*[−/−] enteroids (Supplementary Fig. 11d) and there was no decrease in the levels of glucose, glycerol-3-phosphate, and phosphoenolpyruvate (Fig. 6j, k).

## Purine biosynthesis is dysregulated in the Ankrd9[−/−] intestine

In addition to glycolysis and oxidative phosphorylation, ATP is produced through the de novo synthesis and salvage[3] (Fig. 7a). Very little is known about these pathways in the intestine. IMPDH2 is a rate-limiting enzyme for GTP synthesis within the de novo purine biosynthesis/ salvage pathway. The increased abundance of IMPDH2 in *Ankrd9*[−/−] intestine is consistent with a previously reported role for ANKRD9 in IMPDH2 degradation[10,12]. Upregulation of IMPDH2 can accelerate GTP production at the expense of ATP synthesis (Fig. 7a). Indeed, in *Ankrd9*[−/−] enteroids the adenosine and xanthosine levels (the ATP branch of the pathway) were significantly decreased (by 57% and 19%, respectively, Fig. 7b), and adenine and inosine were lower (by 32% and 43%, respectively, compared to controls, Fig. 7b), although the latter change was not statistically significant. In contrast, the guanine levels were significantly elevated (1.82-fold) and guanosine trended higher (1.46-fold) (Fig. 7b). These results were consistent with the re-

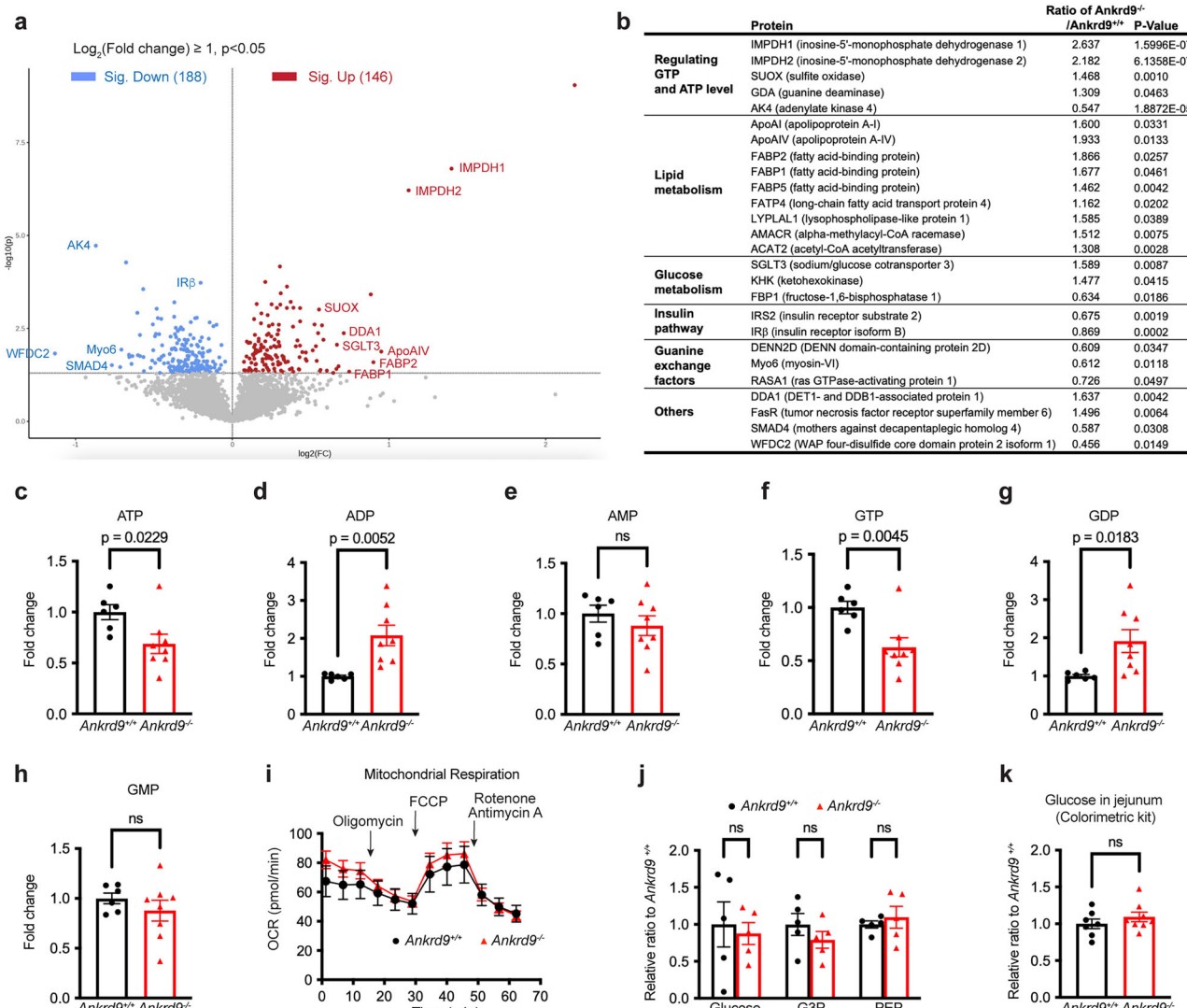

**Fig. 6 | Deletion of Ankrd9 decreases ATP levels without altering mitochondria respiration or glycolysis. a** Volcano plot showing changes in protein abundance in *Ankrd9⁻/⁻* jejunal enteroids compared to *Ankrd9⁺/⁺* enteroids. Proteins that are significantly up- or down-regulated are highlighted in red and blue, respectively. The X-axis shows the fold change and y-axis shows *p*-values. **b** Table lists significantly changed proteins grouped by functional categories. **a, b** $n = 5$ mice per group and 5 individual samples per group; p-values as indicated by a two-tailed unpaired t-test. **c** Levels of ATP, (**d**) ADP, (**e**) AMP, (**f**) GTP, (**g**) GDP and (**h**) GMP in the jejunal enteroids derived from *Ankrd9⁺/⁺* and *Ankrd9⁻/⁻* mice measured by LC-MS; **c**–**h** $n = 6$ and 8 individual samples in *Ankrd9⁺/⁺* and *Ankrd9⁻/⁻* groups, respectively; p-values as

indicated by a two-tailed unpaired t-test; Data are presented as the mean ± SEM. **i** Seahorse analysis of mitochondrial respiration in *Ankrd9⁺/⁺* and *Ankrd9⁻/⁻* enteroids; Data are presented as the mean ± SEM; The data represent three independent experiments. **j** The levels of glucose, glycerol 3-phosphate(G3P) and phosphoenolpyruvate (PEP) in *Ankrd9⁺/⁺* and *Ankrd9⁻/⁻* enteroids measured LC-MS analysis; $n = 5$ individual samples per group; *p*-values as indicated by two-way ANOVA with Šídák test; Data are presented as the mean ± SEM. **k** Glucose level in *Ankrd9⁺/⁺* and *Ankrd9⁻/⁻* jejunum measured by colorimetric kit; $n = 7$ and 8 mice in *Ankrd9⁺/⁺* and *Ankrd9⁻/⁻* groups, respectively; *p*-values as indicated by a two-tailed unpaired t-test; Data are presented as the mean ± SEM.

balancing of nucleotide synthesis from the synthesis of adenine-based purines towards the guanine-based purines.

However, higher IMPDH2 abundance did not result in higher levels of GTP in *Ankrd9⁻/⁻* enteroids; instead, GTP was low when compared to control (Fig. 6f). This result argued against increased IMPDH2 activity and suggested that additional enzymatic steps contributed to purine misbalance. To test this hypothesis, we treated *Ankrd9⁺/⁺* and *Ankrd9⁻/⁻* enteroids with IMPDH2 inhibitor ribavirin. In control cells, inhibition of IMPDH2 with ribavirin increased ATP by 1.37-fold (Fig. 7c). In *Ankrd9⁻/⁻* enteroids, this increase was lower (1.19-fold, Fig. 7c) and insufficient to compensate for the about 30% loss of cellular ATP. Thus, in addition to IMPDH2, other components of the purine biosynthesis/ salvage pathway are affected by Ankrd9 inactivation.

Inosine monophosphate (IMP) is a common precursor of ATP and GTP synthesis at the intersection of purine de novo biosynthesis and

purine salvage (Fig. 7a). IMP levels are significantly lower in *Ankrd9⁻/⁻* enteroids when compared to controls (53% of control, Fig. 7d). The abundance of enzymes involved in purine salvage, such as HPRT and APRT was not changed in *Ankrd9⁻/⁻* cells (Supplementary Fig. 11c). We used AlphaFold3 to predict potential interactions between Ankrd9 and the rate-limiting enzymes within the purine biosynthesis/salvage pathway found an excellent score (ipTM=82) for PRPS1-ANKRD9 complex indicative of likely interactions (Supplementary Fig. 12a). Phosphoribosyl pyrophosphate synthase (PRPS1) generates the key metabolite, phosphoribosyl pyrophosphate, for IMP production and the salvage pathway activity. Consequently, we examined PRPS1 behavior in *Ankrd9⁻/⁻* cells in more detail.

Western blot analysis of total cell lysates found no difference in PRPS1 abundance between control and *Ankrd9⁻/⁻ cells* (Supplementary Fig. 12b), while co-immunoprecipitation of recombinant ANKRD9

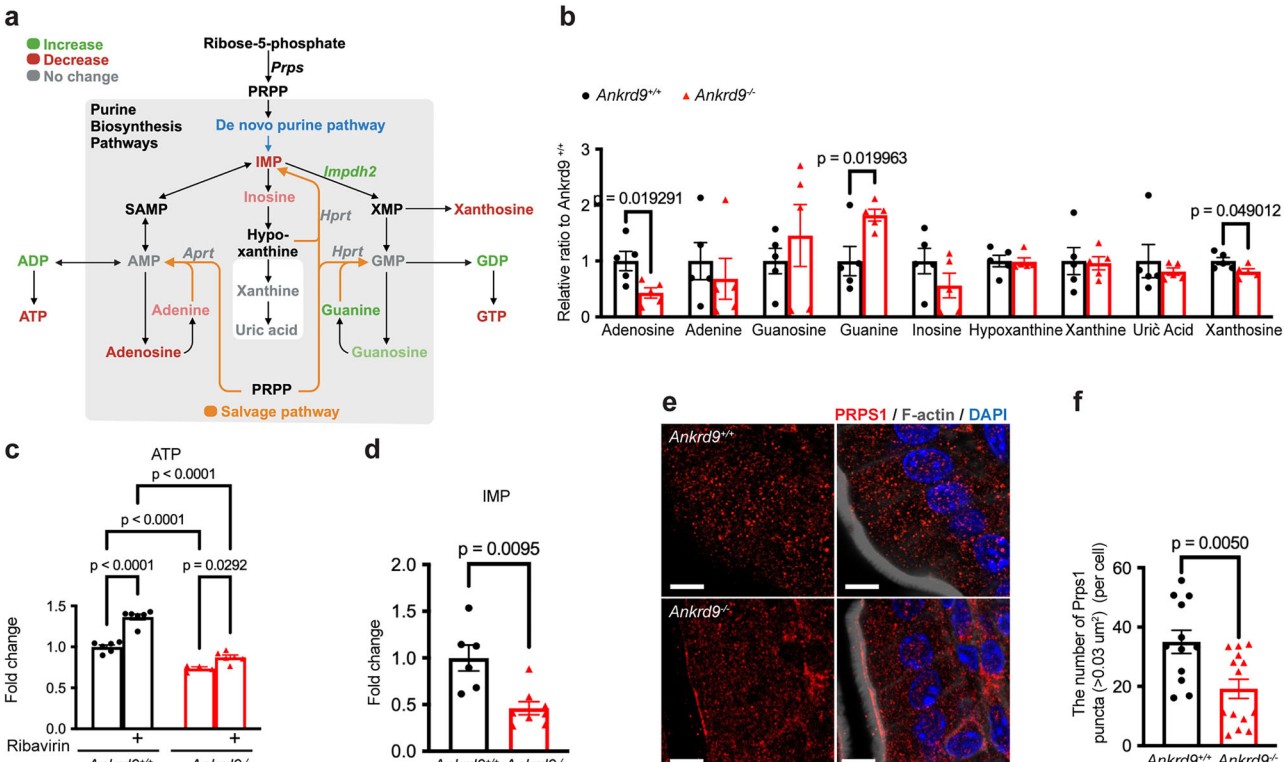

**Fig. 7 | ANKRD9 regulates purine biosynthesis and the salvage pathway at more than one step. a** Schematic of purine biosynthesis pathway (de novo synthesis, salvage and degradation pathway); Created in BioRender. Wang, Y.(2025) https:// BioRender.com/6zq0aa8. PRPP: phosphoribosyl pyrophosphate; PRPS: phosphoribosyl pyrophosphate synthetase; IMP: inosine monophosphate; SAMP: succinyl-AMP; AMP: adenosine monophosphate; ADP: adenosine diphosphate; ATP: adenosine triphosphate; XMP: xanthosine monophosphate; GMP: guanosine monophosphate; GDP: guanosine diphosphate; GTP: guanosine triphosphate. Increased metabolites are in green, decreased metabolites are in red. **b** Nucleosides levels of the purine biosynthesis pathway in jejunal enteroids from $Ankrd9^{+/+}$ and $Ankrd9^{-/-}$ mice analyzed by LC-MS; $n = 5$ individual samples per group; $p$-values as indicated by a two-tailed unpaired t-test; Data are presented as the mean ± SEM. **c** ATP level in jejunal enteroids from $Ankrd9^{+/+}$ and $Ankrd9^{-/-}$ mice with or without 20 μM ribavirin

treatment was determined using luciferase-based assay; $n = 4$–6 individual samples per group, please refer to the Source Data File; $p$-values as indicated by two-way ANOVA with Tukey test; Data are presented as the mean ± SEM. **d** IMP levels in jejunal enteroids derived from $Ankrd9^{+/+}$ and $Ankrd9^{-/-}$ mice analyzed by LC-MS; $n = 6$ and 8 individual samples in $Ankrd9^{+/+}$ and $Ankrd9^{-/-}$ groups, respectively; $p$-values as indicated by a two-tailed unpaired t-test; Data are presented as the mean ± SEM. **e** Immunohistochemical staining of PRPS1(red) in $Ankrd9^{+/+}$ and $Ankrd9^{-/-}$ jejunum; $n = 3$ mice per group per experiment. Data represent three independent experiments; Scale bar: 10 μm. **f** Quantitation of large PRPS1 puncta ($> 0.03$ um²) per cell using Imaris software; $n = 12$–14 sections per group, please refer to the Source Data File; $p$-values as indicated by a two-tailed unpaired t-test; Data are presented as the mean ± SEM.

yielded a specific, but weak PRPS1 signal (Supplementary Fig. 12c). We considered that PRPS1 is an oligomeric protein, whose oligomerization and activity are facilitated by ATP and inhibited by ADP[34–36]. PRPS1 also forms complexes with downstream metabolic enzymes to facilitate substrate channeling[34,37], and is detected in the intestine as large puncta (Fig. 7e). The overall number of PRPS1 puncta in control and $Ankrd9^{-/-}$ enterocytes was not significantly different (Fig. 7e). However, the number of PRPS1 puncta that were larger than 0.03 μm² was significantly lower in the $Ankrd9^{-/-}$ jejunum compared to wild-type controls (Fig. 7f). Thus, in $Ankrd9^{-/-}$ enterocytes the PRPS1's assembly into larger protein complexes was inhibited.

**Nucleotide levels increase during lipid processing in control enterocytes**

Having established the impact of Ankrd9 deletion on chylomicron trafficking and purine synthesis/salvage pathway, we tested whether these two processes were coupled. We reasoned that if chylomicrons trafficking was co-regulated with purine synthesis/salvage, then lipid uptake would affect Ankrd9 and/or components of the purine biosynthesis/salvage pathway. Treating differentiated Caco-2 cells on Transwells with 50 uM oleic acid at the apical side triggered a rapid (within 15 min) change in ANKRD9 pattern. The large puncta in the vicinity of cis-Golgi disappear, the ANKRD9 pattern was reversed at

30 min (Fig. 8). To examine whether lipid uptake impacts purine balance, we took advantage of a well-known property of IMPDH2 to respond to ATP elevation or GTP depletion by forming rod-like structures, called cytoophidia[38], which we confirmed the nucleotide-induced changes in IMPDH2 oligomerization in the wild-type enteroids (Supplementary Fig. 13a).

We then tested how lipid treatment affects IMPDH2. Before treatment of enteroids with oleic acid, IMPDH2 had a fuzzy cytosolic pattern, as expected. Within 15 min of lipid addition, IMPDH2 became clustered and formed puncta and rod-like structures, which persisted at 30 min (Fig. 9a-b and Supplementary Fig. 13b). At 60 min, the IMPDH2 pattern was fuzzy again (Fig. 9a and Supplementary Fig. 13b). In contrast, in $Ankrd9^{-/-}$ enterocytes, IMPDH2 did not form rods in response to lipid uptake despite significantly higher protein abundance (Fig. 9b and Supplementary Fig. 10c). Taken together, these results show that lipid uptake is associated with changes in ATP levels and/or the ATP to GTP ratio and that this response is lost in $Ankrd9^{-/-}$ enterocytes.

To further link purine synthesis to ApoB trafficking, we lowered intracellular ATP by treating Caco-2 cells with 30 uM FCCP and 30 uM oligomycin for 15 min (Supplementary Fig. 14a-b). Confocal sectioning/ imaging of Caco-2 cells from the apical to basolateral side revealed that control Caco-2 cells had significant ApoB signal near the apical aspect

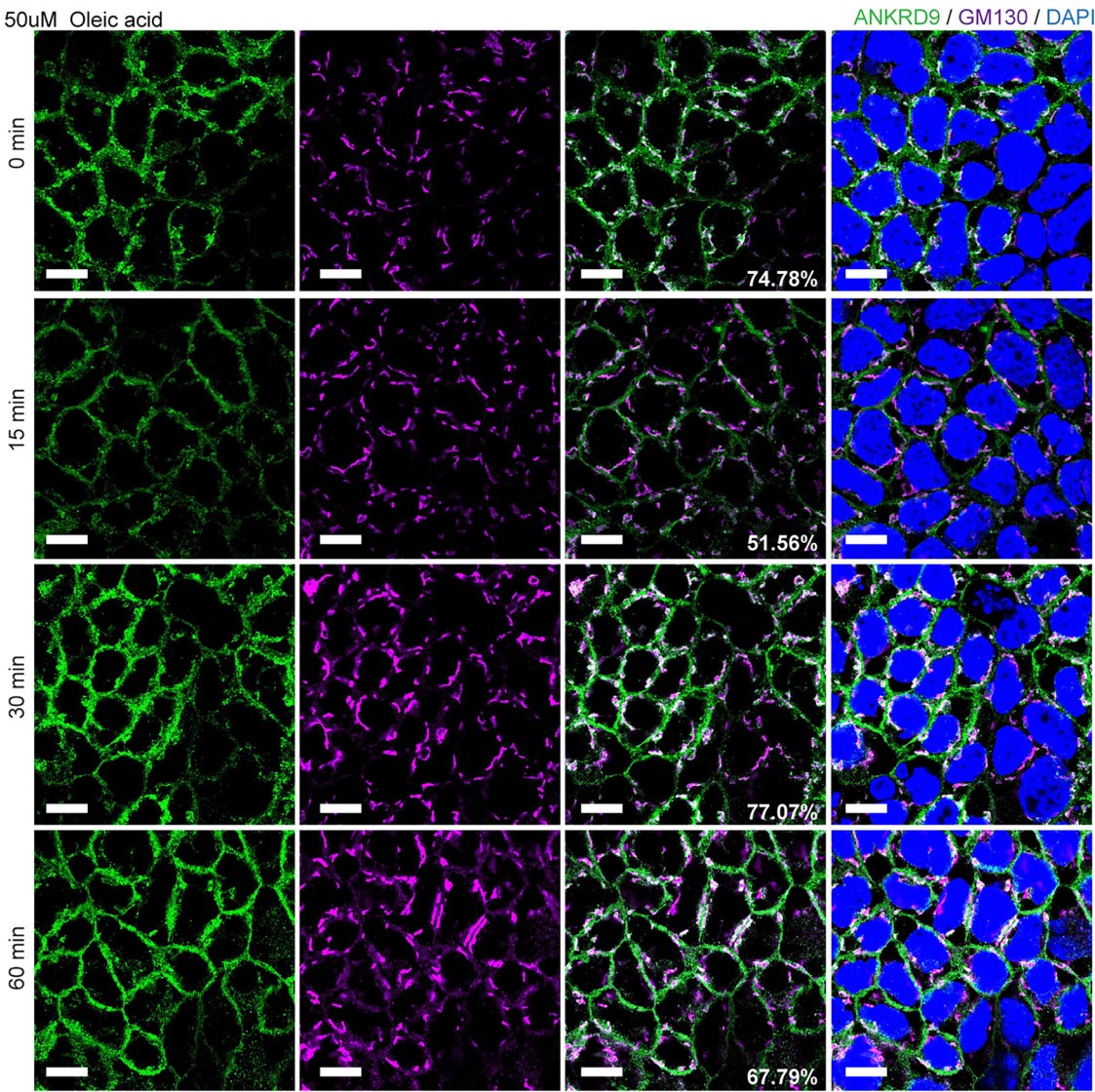

**Fig. 8 | Lipid uptake triggers changes in ANKRD9 intracellular pattern.** Immunofluorescence images of ANKRD9 (green) in differentiated Caco2 cells following treatment with 50 uM oleic acid for 0, 15, 30 and 60 min; Scale bar: 10 μm. The data represent three independent experiments. At steady state (time 0), ANKRD9 shows numerous puncta in the vicinity of cis-Golgi. Within 15 min of lipid uptake, the number of puncta/vesicles near cis-Golgi decreases significantly. At 30 min and further at 60 min, the ANKRD9 pattern returns to one observed at steady state. The proportion of GM130 (purple) co-localizing with ANKRD9 at each time point is indicated by the numbers in the bottom right corner of the image.

of cells, whereas ATP-depleted cells have little or no ApoB at this location. The effect of ATP depletion on ApoB localization was reversed by supplementation of the growth medium with 1 mM ATP for 45 min or by overexpression of recombinant ANKRD9 (Supplementary Fig. 14a-b). Thus, the recombinant ANKRD9 is sufficient to stimulate ATP production necessary for correct ApoB localization.

Finally, we examined whether the role of ANKRD9 in the human intestine can be gleaned from the analysis of several available datasets encompassing (i) results of human single-nuclear sequencing in intestine[39] and (ii) large-scale human variation in intestinal gene expression[40,41]. To understand what processes ANKRD9 expression covaried within intestinal cells and among human subjects, we analyzed genetic correlation structure[42–45] of cellular (Supplementary Fig. 15a) and population ANKRD9 (Supplementary Fig. 15b). Analyses of the top gene-set enrichment (GSEA) terms arising from ANKRD9-correlated genes showed pathways enriched for metabolic substrate transport and fatty acid metabolism in both cells and population variation. These data support a conserved role for ANKRD9 in lipid metabolism transport processes in mice and humans.

## Discussion

Every meal triggers a cascade of energy-requiring steps to drive the uptake, packaging, and release of nutrients to the rest of the body. We demonstrate that purine biosynthesis/salvage in the small intestine plays an important role in supplying ATP necessary for the efficient trafficking of lipids and ApoB, the major components of chylomicrons. We also show that a highly conserved protein ANKRD9 plays a central role in coupling purine biosynthesis to ApoB trafficking. Without ANKRD9, purine biosynthesis/salvage pathway is dysregulated, leading to low levels of ATP (and GTP) and delayed triglyceride synthesis/ApoB trafficking. The localization of ANKRD9 in the vicinity of cis-Golgi and the lateral membrane and changes in cis-Golgi morphology in the absence of ANKRD9 suggest that ANKRD9 may be particularly important in regulating local nucleotide pools.

Previously, the central role of mitochondria in dietary lipid transport was proposed[46]. The marked effect of mitochondrial disruption on intestinal morphology and function[46] does support the essential role of respiration for normal functions of the intestine, including lipid transport. However, it remained unclear whether

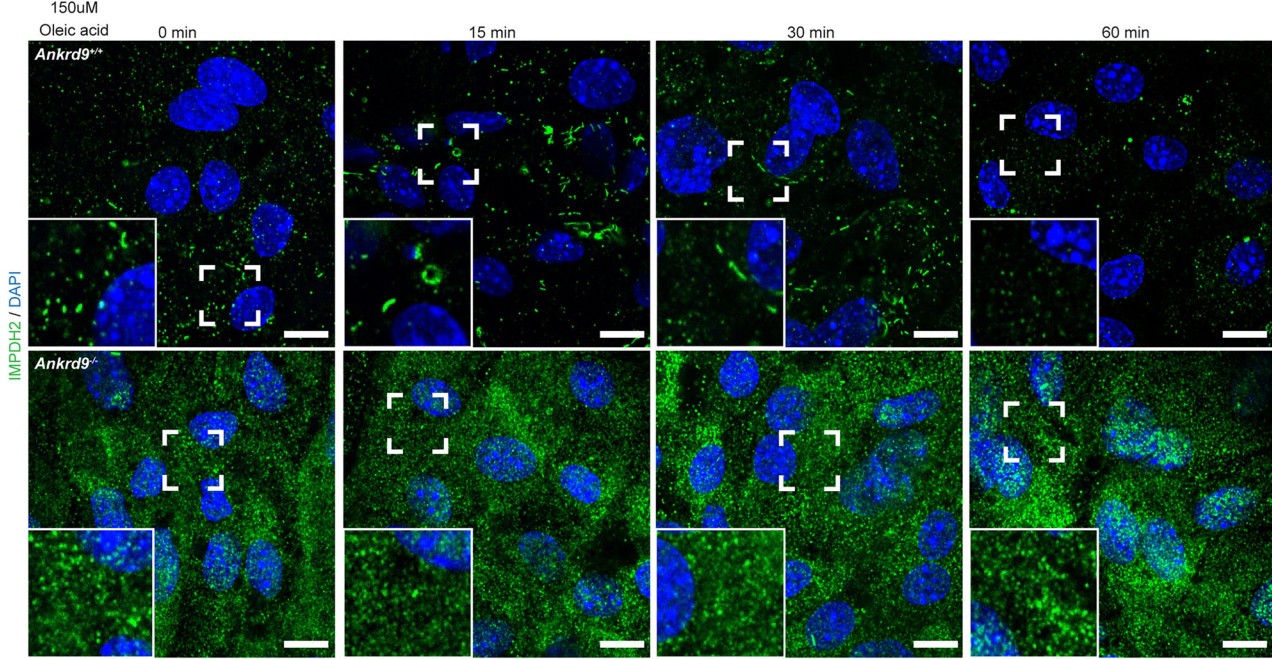

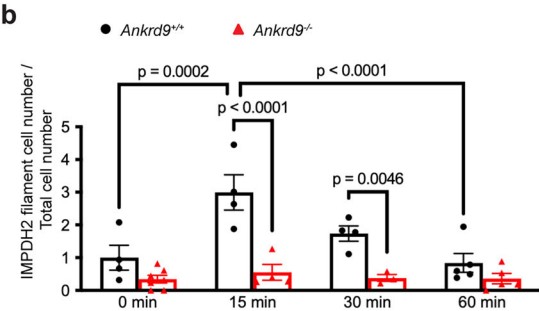

**Fig. 9 | Lipid uptake increases ATP levels in *Ankrd9*⁺/⁺ but not in *Ankrd9*⁻/⁻ cells.** **a** Immunostaining of IMPDH2 (green) in differentiated jejunal *Ankrd9*⁺/⁺ and *Ankrd9*⁻/⁻ enteroid monolayers after treatment with 150 uM oleic acid for 0, 15, 30 and 60 min; The bounding boxes indicated the magnified regions at the lower left corner of each image. Scale bar: 10 μm; Data represent three independent experiments. **b** The quantification of IMPDH2 filaments per cell; $n = 3–7$ sections per group, please refer to the Source Data File; $p$-values as indicated by two-way ANOVA with Tukey test; Data are presented as the mean ± SEM.

mitochondria are responsible for an increase in ATP needed when nutrients enter the enterocytes. Our data show that this specific function belongs to purine biosynthesis and salvage pathway, which is upregulated in response to lipid entry. Inactivation of Ankrd9 disregulates this pathway along with lipid processing, without altering mitochondria function and without negatively affecting intestinal morphology. We propose that purine biosynthesis/salvage pathway enables the intestine to meet high energy demands during nutrient trafficking and supplements energy derived from mitochondrial respiration and glycolysis. Our data also suggests that the multipronged role of ANKRD9 in this process is determined by its ability to form distinct spatially separated protein complexes: the 440 kDa complex associated with membranes and smaller protein complexes in the cytosol. The membrane-bound complex is lipid responsive and changes its location near cis-Golgi when trafficking of ApoB is initiated.

Our data helps to reconcile previous disparate and inconsistent reports about the ANKRD9 function. Upregulation of ANKRD9 in response to energy deficit (observed in cells with disrupted fatty acid oxidation[9]) would increase ATP production and could be a beneficial compensatory response. The binding of ANKRD9 to endomembranes and the effect on Golgi morphology may explain the previous report of ANKRD9-dependent changes in the cellular copper[13] because the Golgi-located copper transporters play a central role in the regulation of copper homeostasis[13]. The precise mechanism of ANKRD9-dependent dysregulation of copper balance remains to be established.

Lastly, animals with inactivated Ankrd9 show a significant decrease in body fat on a regular diet. Based on our findings, the lean body phenotype observed in *Ankrd9*⁻/⁻ mice could be explained by (i) slow fat processing in the intestine, resulting in intestinal fat retention; (ii) high abundance of Ankrd9 in the heart and skeletal muscle, where loss of Ankrd9 may cause energy deficits similar to intestine. Finally, although deletion of Ankrd9 did not affect random blood glucose levels (Supplementary Fig. 2c), it significantly reduced fasting (16 h) blood glucose levels (Supplementary Fig. 2d). Together, the results suggest that *Ankrd9*⁻/⁻ mice may utilize body fat more efficiently as an energy source to sustain physiological processes under fasting conditions. Going forward, it would be interesting to see whether Ankrd9 has the same protective effect against a high-fat diet and whether transient manipulation of Ankrd9 levels or complex formation during nutrient intake can provide control of dietary fat absorption.

## Methods

### Animal experiments

All animal protocols were approved by the Institutional Animal Care and Use Committee of the Johns Hopkins University (JHU ACUC, protocol #MO17M385, MO20M333 and MO23M337). The mice were maintained under specific pathogen-free (SPF) conditions with a controlled temperature of 22–24 °C, relative humidity of 40–60%, and a 14 h light / 10 h dark cycle, with free access to water and food unless otherwise specified.

*Ankrd9*[−/−] mice on C57BL/6N-*A*[tm1Brd] background were obtained from the Mutant Mouse Resource and Research Center (MMRRC) at the University of California at Davis (UC Davis, C57BL/6N-*A*[tm1Brd] *Ankrd9*[tm1(KOMP)Wtsi]Mmucd, RRID:MMRRC _046599-UCD). In these mice, exon 3 of *Ankrd9* was replaced by FRT- LacZ - loxp - neomycin resistance (Neo) cassette; this resulted in the loss of Ankrd9 expression and the expression of LacZ under the endogenous Ankrd9 promoter. *Ankrd9*[−/−] and *Ankrd9*[+/+] littermate controls were generated by heterozygous *Ankrd9*[+/-] breeding and were maintained on C57BL/6 N background. Fourteen weeks old homozygous Ankrd9 knockout male mice (*Ankrd9*[−/−]) and their wild-type (*Ankrd9*[+/+]) male littermates were used in the experiments conducted for this study. For *Ankrd9*[−/−] mice genotyping, the forward primer: 5′-GGGATCTCATGCTG-GAGTTCTTCG; and the reverse primer: 5′-AGTCTGGCTCCCATTCAA-CACC were used. For the genotyping of wild-type mice, the forward primer was 5′-GAGTGAATACCTGCATCTACAGAACCC and the reverse primer was 5′-TGTCCTCGAGTAGCCAAACTGGC.

### Measurement of body composition by magnetic resonance imaging (MRI)

The body composition was assessed in 14-week-old male mice using MRI scanning (EchoMRI, Houston, TX, USA) conducted by the Johns Hopkins University Phenotyping and Pathology Core. Four components: lean mass, fat mass, free water, and total water were measured.

### Enteroid culture

Enteroids were isolated from adult mouse jejunum (14-week-old male mice, at least 5 male mice per group). All procedures were conducted under sterile conditions within a tissue culture hood. Briefly, the intestines were harvested and washed in ice-cold sterile CCS buffer (5.6 mM $Na_2HPO_4$-$2H_2O$, 8 mM $KH_2PO_4$, 95 mM NaCl, 1.6 mM KCl, 44 mM sucrose, 55 mM D-sorbitol). The tissue was minced and treated with 120 mM EDTA for 1 h, followed by passage through a 70 μm cell strainer. Crypt cells were subsequently isolated by centrifugation at $200 \times g$ for 10 min at 4 °C, then washed with growth media from the Johns Hopkins University Organoid Core[27]. Isolated crypt cells were suspended in Matrigel (Corning, Corning, 356231, NY) at 100–300 cells per 30 ul Matrigel in 24-well plastic plates with 500 ul of growth medium. For 3D enteroid differentiation, passaged enteroids were cultured in growth medium for 2 days, and then switched to differentiation medium[27] and cultured for 48 h at 37 °C in a 5% $CO_2$ incubator. For monolayer enteroid differentiation, the passaged enteroids were dissociated into single cells by pipetting and cultured with growth medium for two days in a Transwell (Corning, #3470). Subsequently, the medium was replaced with differentiation medium, and the cells were cultured for an additional 72 h at 37 °C in a 5% CO2 incubator. Then enteroids were collected for the following experiments. Refer to figure legends for specific treatment conditions. Human jejunal enteroids were provided by Dr. Jennifer Foulke-Abel from the Department of Gastroenterology, Johns Hopkins University School of Medicine.

### Cell lines

Caco-2 cells were obtained from Dr. Mark Donowitz (Department of Gastroenterology, Johns Hopkins University School of Medicine)[29] and were cultured with growth medium (DMEM with 10% FBS) at 37 °C with 5% $CO_2$. The cells were grown in a T75 flask for initial experiments. Caco-2 cells were cultured in Transwell (Corning, #3470) for 14 days. For the lipid treatment experiment, cells were cultured in an FBS-free DMEM medium for 2 h. Then FBS-free DMEM medium was added to the basolateral chamber, and FBS-free DMEM medium containing 50 uM oleic acid (Sigma, O1257) in FBS-free DMEM medium was added to the apical chamber and cells were incubated for 0, 15, 30 and 60 min. Then cells were fixed by 4% paraformaldehyde (PFA) for subsequent immunostaining experiments.

### Olive oil treatment

Fourteen weeks old male mice were fasted for 6 h prior to oral olive oil treatment. Olive oil (10 ml/kg body weight, Sigma, O1514) was then orally administered by gavage. Mice were sacrificed at the following time points: 0 (fasting levels), 15, 30, 60, 90, and 120 min and jejunum was harvested for the next experiments, with a sample size of 3–5 mice per group for each time point, please refer to the Source Data file.

### Cryosection

Jejunums were washed with ice-cold PBS and fixed in 4% PFA for 6 h at 4 °C. The samples were then incubated in 30% sucrose overnight before being embedded in Optimal Cutting Temperature compound (OCT, Sakura Fintek). Ten μm cryosections were made for subsequent staining. Enteroids were treated with 20 uM Ribavirin (Sigma, R9644) or 0.5 mM ATP (Sigma, A6419) in differentiation medium for 4 h, then enteroids were fixed in 4% PFA for 30 min at room temperature and incubated in 30% sucrose for 1 h before being embedded in OCT.

### Immunohistochemistry

Cryosections were washed in PBS, then incubated with blocking buffer (5% Bovine Serum Albumin (BSA)) with 0.3% TritonX-100 for 1 h at room temperature. The sections were incubated with the primary antibodies (please refer to the antibody list in the Supplementary Table 1 for details) overnight at 4 °C. The secondary antibodies were diluted at 1:1000 and incubated for 1 h. Lipids and lipid-containing compartments were stained with BODIPY™ 500/510 C1,C12 (Thermo Fisher Scientific, D3823). F-actin was stained by Alexa Fluor 647 phalloidin (Thermo Fisher Scientific, A22287). All the slides were mounted by the ProLong Gold Antifade Mountant with DAPI (Thermo Fisher Scientific, P36935). Images were acquired using Zeiss Confocal LSM 800. Five Z stacks were acquired and then merged using Zeiss Zen software.

### Electron microscopy (EM)

The EM study on the jejunum of 14-week-old male *Ankrd9*[+/+] and *Ankrd9*[−/−] mice. Jejunum tissues (2–3 cm in length) were harvested from each mouse and minced into small pieces (approximately 1–2 mm³) using curved surgical scissors to ensure proper infiltration of the fixation buffer. We prepared 10–20 biopsies per mouse, and the samples were then processed at Johns Hopkins Institute for Basic Biomedical Science Microscope Facility. Jejunum was fixed by immersion with freshly prepared EM grade fixative (2% paraformaldehyde, 2% glutaraldehyde and 5 mM $MgCl_2$ in 0.1 M Sorenson's phosphate buffer, pH 7.4) at 4 °C overnight. Samples were rinsed with a wash buffer-1 (3% sucrose, 1.5% potassium ferrocyanide, 2% reduced osmium tetroxide in 0.1 M Sorenson's phosphate buffer) for 2 h at 4 °C. Samples were then rinsed by wash buffer-2 (3% sucrose in 0.1 M maleate buffer, pH 6.2), and en-bloc stained by 2% uranyl acetate in 0.1 M maleate buffer at 4 °C dark for 1 h. Samples were then dehydrated in a series of graded ethanol, brought to room temperature in the 70% ethanol step and completely dehydrated in 100% ethanol. Samples were resin embedded (Epon 812, T. Pella) after a propylene oxide transition step, and further infiltrated and cured the next day. 60 nm thin sections were obtained with a Diatome diamond knife (45 °C). Four to five pieces were cut out of jejunum tissue to prepare grids.

Then, 2-3 sections per grid were analyzed. Sections were picked up on 1 × 2 mm formvar coated copper slot grids (Polysciences) and further stained with uranyl acetate followed by lead citrate. Grids were examined on a Hitachi H-7600 TEM operating at 80 Kv. Images were digitally captured with an AMT XR 80-8-megapixel CCD camera; 25−35 images per mouse were captured and Golgi size was quantified for 80−90 cells in each group. All the EM procedures were performed by the Johns Hopkins Institute for Basic Biomedical Science Microscope Facility.

## Proteomics

Comparative analysis of proteomes was performed by the Johns Hopkins Mass Spectrometry and Proteomics Core. Enteroids were collected after 48 hours of culture with a differentiation medium, and proteins were extracted. Two groups (*Ankrd9*[+/+] and *Ankrd9*[−/−] group, five biological replicates per group) and 100 ug protein per each sample were used for mass-spectrometry analysis. The initial stage of sample preparation involved reducing the proteins with 2.5 uL of 15 mg/mL Dithiothreitol (DTT) in 100 mM triethylammonium bicarbonate (TEAB) for one hour at 57 °C with agitation. Subsequently, the pH was adjusted to 8.0 using 500 mM TEAB, followed by alkylation with 2.5 µl of 250 mM Methylthiol for ten min at ambient temperature. On ice, an amount equivalent to eight times the volume of TCA/acetone was added, and proteins were precipitated at −20 °C overnight. Protein pellets (less than 100 ug) were washed twice with cold acetone, dried, and reconstituted in 95 ul of 100 mM TEAB containing 5% acetonitrile. The mixture was then sonicated for twenty min, subsequently digested with 3.5 ug of Trypsin (Pierce) at 37 °C overnight.

Tandem Mass Tag (TMT) labeling and mass spectrometry have been documented previously[47,48]. Peptides were labeled with Isobaric Mass Tags TMT16 Pro reagents in 20 µl of acetonitrile following the manufacturer's protocol. Specifically, peptides from each sample were reconstituted in 100 µl of 100 mM TEAB buffer and then labeled with a unique TMT 10-plex reagent (Thermo Fisher Scientific, 90110, Lot: VG306772) in 40 µl acetonitrile, using the entire reagent volume for each sample. An additional 8 ul of 5% hydroxylamine was added to quench the reaction, followed by a 15 min incubation. The TMT-labeled peptides from all samples were then combined to yield a total of 1000 ug of labeled peptides. This mixture was divided into three approximately 333 µg aliquots and dried.

Fractionation using basic reverse phase (bRP) chromatography was performed on a single aliquot containing 333 ug of combined TMT-labeled peptides. The sample was dried, reconstituted in 50 ul of 50 mM TEAB buffer, and purified with Pierce detergent removal columns to remove excess labels, small molecules, lipids, and other contaminants, resulting in an elution volume of 50 ul TEAB buffer. To this, 1950 ul of bRP buffer A−containing 10 mM TEAB in water−was added, and 2 µl of the peptide solution was injected over 8 min at a flow rate of 250 ul/min. Peptides were then separated using basic reverse phase chromatography with an 85-minute gradient from 100% solvent A (10 mM TEAB in water) to 100% solvent B (90% acetonitrile with 10 mM TEAB) at 250 µl/min on an XBridge C18 column, 5 µm particles, 2.1 ×100 mm (Waters), equipped with a 5 µm guard column (2.1 × 10 mm, Waters). The system was an Agilent 1200 series HPLC with binary capillary pumps, a variable wavelength UV detector, and a micro-fraction collector. The resulting 84 fractions were combined into 24 for LC-MS/MS analysis.

LC-MS/MS analysis: Peptides in fraction 1, with a calculated injected amount of 400 ng, exhibited a low-intensity base peak chromatogram. An injection of 1040 ng yielded an intensity of $2 \times 10^9$. All other fractions were analyzed with a calculated injected amount of 1040 ng. Peptides from 24 fractions, averaging 13.9 ug each, were reconstituted in 80 µl of 2% acetonitrile/0.1% formic acid, and 6 ul of the calculated average amount per fraction (1040 ng) was analyzed via nano-LC-MS/MS. The peptides were examined using a nano-LC-Orbitrap-Lumos-ETD system (Thermo Fisher Scientific) coupled to an EasyLC1200 series. Separation employed reverse-phase chromatography with a gradient from 2% to 90% acetonitrile / 0.1% formic acid over 100 min at a flow rate of 300 nl/min on a 75 µm × 150 mm ProntoSIL-120-5-C18 H column, 3 µm particle size, 120 Å pore size (BISCHOFF). Eluted peptides were introduced into the Orbitrap-Lumos-Fusion mass spectrometer through a 1 µm emitter tip (New Objective) at 2.4 kV. Full MS survey scans were collected within an m/z range of 375-1600 on the Orbitrap (Orbi-trap), using a data-dependent Top 15 method with a 15-second dynamic exclusion. Precursor ions were isolated with a 0.7 Da window and fragmented via higher-energy collisional dissociation (HCD) at a collision energy of 38. Precursor and fragment ions were analyzed at resolutions of 120,000 and 60,000, respectively, with parameters tailored to the requirements of the TMT experiment.

For MS/MS data analysis, Tandem MS2 spectra (signal/noise >2) were processed using Proteome Discoverer (version 2.4, Thermo Fisher Scientific), employing the Files RC option for recalibration with an appropriate database. The MS/MS spectra were searched with Mascot version 2.8.1 (Matrix Science, London, UK) against the RefSeq2021_204_mus_musculus database, as well as a supplementary database containing enzymes and BSA. Trypsin was used as the enzyme, allowing a maximum of two missed cleavages. The precursor mass tolerance was set at 5 ppm, and the fragment mass tolerance at 0.01 Da. TMT6 labeling was applied to the N-terminus and lysine residues as fixed modifications. Variable modifications included methionine oxidation and deamidation on asparagine and glutamine residues. Peptide identifications obtained from Mascot were processed through Proteome Discoverer and Percolator to identify peptides with a confidence threshold of 1% False Discovery Rate (FDR), based on an auto-concatenated decoy database search, as well as to compute protein and peptide ratios. Only peptides with Rank 1 were considered for analysis. Normalization and ratio calculations were performed exclusively on unmodified peptides. The criteria for data inclusion comprised precursor signal-to-noise ratio greater than 2, reporter signal-to-noise ratio greater than 5, and an isolation interference of less than 25%. Database searching against RefSeq2021_204_mus_musculus identified a total of 6376 proteins across high, medium, and low confidence levels, with at least one peptide identified at a 1% FDR and Rank 1 (Percolator FDR).

## Measurements of nucleotide levels

Differentiated mouse enteroids were collected from two groups (*Ankrd9*[+/+] and *Ankrd9*[−/−] group, 6−8 biological replicates per group; please refer to the Source Data File). The samples were homogenized in 200 ul of 1 N cold perchloric acid, subjected to sonication for 80 seconds, and subsequently centrifuged at 4 °C at 13,000 × g for 2 min. The supernatant was then transferred into a new Eppendorf tube. The pH was subsequently adjusted to 7.0 using 1 M $K_2CO_3$, followed by a second centrifugation conducted at 4 °C at 13,000 × g for a duration of 2 min. The resulting supernatant was utilized for the quantification of protein content, intracellular ATP and other nucleotide levels, employing either the luciferase-luciferin system (ATP determination kit, Thermo Fisher Scientific, A22066) or high-resolution mass spectrometry (HRMS) analysis[49,50]. Thermo Vanquish Ultra-high performance liquid chromatography system coupled with Q Exactive Hybrid Quadrupole-Orbitrap Mass Spectrometer (UPLC-QE Orbitrap MS) equipped with a heated electrospray ionization probe (Thermo Fisher, CA, USA) was used. An Xbridge BEH Amide 2.5 µm 2.1 × 150 mm column (Waters Corp, Milford, MA, USA) was applied for nucleotide separation in negative ionization modes with separate injections. Mobile phase A was 10 mM ammonium acetate (pH = 8.5), and mobile phase B was acetonitrile/$H_2O$ = 90/10 (v/v), containing 10 mM ammonium acetate. A linear gradient elution program was set as 90% B from 0 to 0.1 min, decreasing to 48% B from 0.1 to 12 min for

1 min; the mobile phase composition was then returned to 90% B from 13 to 13.1 min for 2.9 min. The total run time was 16 min. The flow rate was 0.40 ml/min and the column temperature was at 10 °C. The resolution for data collection in the full scan was 70,000 in the ranges of m/z 300–900. The automatic gain control (AGC) target was 3e6 and the maximum IT was 100 ms. The targeted-SIM mode parameters were set as a resolution at 70000, AGC target at 5e4, maximum IT of 100 ms, and isolation window of 1.0 m/z.

Raw LC-MS data were processed in Quan Browser (version 4.3.73.11) using batch reprocessing. A predefined processing method ("Proc Meth") was applied to extract peak areas of targeted nucleotides. Each chromatographic peak was visually inspected to verify correct identification, especially in cases of retention time shifts caused by column performance variability. Manual integration was performed when needed to ensure accurate quantification. Verified peak areas were then exported for downstream analysis. Absolute quantification of nucleotides was achieved using external calibration curves generated from commercial analytical standards across defined concentration ranges, with concentrations in biological samples calculated based on the corresponding linear regression equations. The results were normalized with protein amount.

### Seahorse analysis

Thirty enteroids were suspended in 2 ul Matrigel in a collagen-coated XFe96 well Spheroid Microplate with 100 ul growth medium. After 48 h culturing, the medium was switched to a differentiation medium for 48 h. Then the medium was changed to seahorse assay medium (pH7.4), including 2 mM L-Glutamine, 1 mM pyruvate and 10 mM glucose in XF Base medium (Agilent, 102353-100). Mitochondrial respiration was measured by Seahorse XFe96 extracellular Flux analyzers under treatment of oligomycin A (2.5 uM), carbonyl cyanide-4-phenylhydrazone (FCCP, 2.5 uM), and rotenone (2.5 uM) along with antimycin A (2.5 uM). The oxygen consumption rate was normalized to protein amount.

### Metabolite measurements

Differentiated mouse enteroids were collected from two groups (*Ankrd9*$^{+/+}$ and *Ankrd9*$^{-/-}$ group, 5 biological replicates per group; please refer to the Source Data File). The samples were homogenized in 200 ul methanol using a Beadbeater (MiniBeater-16, Model 507). The mixture was then sonicated in ice water for 30 min and left at −20 °C for 20 min. After centrifugation at $10,000 \times g$ and 4 °C for 10 min, 180 ul of supernatant was used for analysis. A Thermo Vanquish UPLC system coupled with a Q-Exactive Orbitrap mass spectrometer equipped with a heated electrospray ionization (HESI) probe (Thermo Fisher, CA, USA) was used in this study. An Xbridge BEH Amide 2.5 μm 2.1 × 150 mm column (Waters Corp, Milford, MA, USA) was applied for polar separation in both negative and positive ionization modes with separate injections. Mobile phase A was acetonitrile/$H_2O$ = 10/90 (v/v), containing 5 mM ammonium acetate and 0.1% acetic acid, and mobile phase B was acetonitrile/$H_2O$ = 90/10 (v/v), containing 5 mM ammonium acetate and 0.1% acetic acid. A linear gradient elution program was set as 70% B from 0 to 0.1 min, decreasing to 30% B from 0.1 to 5 min and held for 4 min; the mobile phase composition was then returned to 70% B by 2 min and held for 9 min. The total run time was 20 min. The flow rate was 0.30 ml/min, and the column temperature was 40 °C. The automatic gain control (AGC) target was 3e6 and the maximum IT was 200 ms. The results were normalized with protein amount.

For the metabolomics analysis, the Compound Discover software (version 3.3, Thermo Fisher Scientific) was employed to process and analyze the '.raw' data files obtained from the UPLC-QE Orbitrap MS. To initiate the data processing task, a new study was created, and a customized non-targeted metabolomics workflow was employed for data processing. The '.raw' data files were added to the project

and categorized into three predefined groups: Blank, Samples, and Identification only. The identification of compounds was performed using mzCloud (ddMS2), ChemSpider (formula or exact mass), and an in-house database containing m/z values of 171 standards and 20 $^{13}C^{15}N$ amino acids. The workflow included retention time correction, feature detection, and chromatogram alignment. The parameters used were as follows: polarity (positive [M + H]$^+$ or negative [M-H]$^-$) determined by the raw data, maximum shift of 0.2 min, mass tolerance of 5 ppm, and a minimal peak intensity of $1 \times 10^4$. After data processing, a characteristic table was generated based on the m/z and retention time of each molecule, which provided the peak areas of each compound across all samples. Subsequently, the data was normalized and exported in.csv format. Quality control (QC) samples were utilized to calculate the coefficient of variation (CV) for each compound, and compounds with CV values less than 20% were selected for further statistical analysis[51].

### Statistical analysis

Statistical analyses were conducted utilizing GraphPad Prism. Significance was established at the level of $p < 0.05$, and all data presented in the figures are expressed as the mean ± SEM. A minimum of three samples per group were chosen for statistically meaningful interpretation of results and differences in the studies using the two-tailed Student's t-test and analysis of variance (ANOVA).

### Reporting summary

Further information on research design is available in the Nature Portfolio Reporting Summary linked to this article.

## Data availability

All the necessary data supporting the conclusions of this research are included within the manuscript, supplementary information, and Source Data files. The raw proteomic data have been deposited to the ProteomeXchange Consortium (https://www.ebi.ac.uk/pride) via the PRIDE partner repository with the dataset identifier PXD073421, with the project title "The role of ANKRD9 in mouse jejunal enteroids". The raw data pertaining to lipidomics, metabolomics, and nucleotide measurements have been deposited in the MassIVE repository, with accession numbers MSV000100414 for lipidomics, MSV000100413 for metabolomics, and MSV000100415 for nucleotide measurements. The Source Data file accompanying this publication includes data from individual experiments utilized to derive the final figures. These data are provided as a Source Data file accompanying this publication. Source data are provided with this paper.

## Code availability

We have developed custom code for the ANKRD9-correlation analysis derived from human single-nuclear sequencing data, which has been uploaded to GitHub: https://github.com/mingqizh/ANKRD9-Intestine-Integration/tree/main.

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

## Acknowledgements

This work was entirely funded by the National Institutes of Health (NIH) under grant R01DK071865 awarded to S.L., with partial support from NIH grant R35GM133510 to J.Z., and also supported by NIH grants DP1DK130640 and U54OD039864 to M.S. We thank Dr. Mark Donowitz of Johns Hopkins University School of Medicine for sharing the Caco2 cells, Dr. Jennifer Foulke-Abel of Johns Hopkins University School of Medicine for sharing the human enteroids, Dr. Nicholas Zachos of Johns Hopkins University School of Medicine for advice to enteroid culture, Dr. Edward C. Twomey of Johns Hopkins University School of Medicine for

helping with the protein complex analysis, Mr. Michael Delannoy of Johns Hopkins Institute for Basic Biomedical Science Microscope Facility for EM sample preparation, Dr. Taekyung Ryu of Johns Hopkins University School of Medicine for uploading the proteomes data to the ProteomeXchange, and all members of Dr. Lutsenko laboratory for useful comments.

## Author contributions

Conceptualization, Y.W. and S.L.; Methodology, Y.W., L.C., Y.M., M.Z., A.G., M.S., J.Z., A.M.Z., X.D., R.N.C., and S.L.; Investigation, Y.W., L.C., Y.M., M.Z., M.S., J.Z., R.N.C., and S.L.; Visualization, Y.W., L.C., Y.M., M.Z., M.S., J.Z., and S.L.; Writing-Original Draft, Y.W. and S.L.; Funding Acquisition, S.L.; All authors contributed to editing and reviewing the manuscript.

## Competing interests

The authors declare no competing interests.
