## [Transparent Peer Review file · Nature Communications]

Enterocytes Rely On Purine Biosynthesis/Salvage Pathway To Facilitate Dietary Fat Absorption

Corresponding Author: Professor Svetlana Lutsenko

Version 0:

Reviewer comments:

Reviewer #1

(Remarks to the Author)

Critique: This is potentially an interesting topic which is not very well understood. Despite my enthusiasm for the topic, I have many problems with the paper which greatly diminished my enthusiasm for the paper. I have summarized my concerns below:

1. This is a very complex paper to read and to understand. I have read the manuscript carefully a couple of times, but I still have difficulty of being convinced of the conclusions made by the authors. The authors interchangeably use the term of fat absorption and nutrient absorption. They are two very different terms referring to two totally different functions carried out by the gastrointestinal tract. There are three major macronutrients in the nutrients processed by the small intestine, lipid, protein and carbohydrate. Thus it is imperative that this is clearly defined in the manuscript. Within the process of intestinal fat absorption, there are also three major steps, uptake, resynthesis of triglyceride, and the formation and secretion of chylomicrons. By Thus this paper has to be totally rewritten before it should be reviewed by any journal.

2. When the authors are discussing intestinal fat absorption, they should clearly define which component of intestinal fat absorption they are talking about. Some concepts in the paper are not accurate. For instance, the authors referred to the uptake of fatty acids by the small intestinal epithelial cells as being active. This is not correct. Most investigators in the field of fat absorption would consider the initial uptake of fatty acids as being passive and is driven by the concentration gradient of fatty acids between the lumen and the intracellular compartment.

3. Histology (immunohistochemistry) and electron microscopy are major techniques utilized in this study. While these are accepted tools in many biological and physiological research, an important consideration is the possibility of misinterpretation because of the small number of samples studied. For instance, how many sections are studied in each animal and the statistical analysis are obviously important considerations. Just studying 3 animals per group is not rigorous enough for one to draw conclusions about an observation. This important question is not addressed at all in the manuscript. This is a major weakness of the manuscript.

4. In figure 2, the investigators harvest the jejunal tissue for histology at different time points after the oral administration of olive oil. As pointed out above in 3, I am not sure if the fluorescence intensity data is totally accurate without knowing how many tissue sections were examined in each animal. How reproducible is the observation from the wild type and the knockout animals. Also, it is a real pity that the investigator did not harvest any plasma samples in the animals to measure triglyceride level. This would provide them with a good measure of the amount of lipids transported by the gut at different time points after oleic acid feeding.

Furthermore, most investigators studying intestinal lipid absorption and transport monitor the plasma lipid levels for 4 hours. How can the investigators be sure that 1 hour was the ideal length of the study. The fluorescence was still going up between 30 – 60 mins in the knockout animals while the while the WT animals was decreasing between 30 – 60 mins. This is a major deficiency in experimental design.

5. The quality of the histology and immunohistochemical images were not particularly good. For instance, figure 3 of the apoB48 staining was not particularly convincing. I am not convinced that one can conclude with confidence what the images are telling us about a particular physiological process.

6. The apo B 48 data should be interpreted with caution. Yes, apo B with associated with chylomicrons and there is one apo B per chylomicron particle. It has been demonstrated elegantly by Davidson and Glickman from Columbia University apo B secretion is not altered by the amount of lipid secreted in chylomicrons. This observation has been confirmed by other investigators using the lymph fistula rat model. Thus the movement of apo B should not be interpreted as chylomicron formation or secretion. The only way to directly quantitate the amount of chylomicron triglyceride secreted is by the lymph fistula mouse model. Though technically difficult, it is certainly doable.

Reviewer #3

(Remarks to the Author)

The study by Wang and colleagues identified ANKRD9 as an important regulator of dietary lipid metabolism in the mouse intestine. They started with the observations that Ankrd9 KO jejunum had robust lipid accumulation compared with the controls. They demonstrated that ANKRD9 was concentrated near the cis-Golgi and promoted APOB transport through the Golgi in response to lipid uptake. Interestingly, they found that ANKRD9 also regulated nucleotide metabolism, and that Ankrd9 KO jejunum/enteroids had lower levels of ATP, GTP and IMP. The authors further showed that lipid loading increased intracellular ATP levels in an ANKRD9-dependent manner, and that intracellular ATP levels conferred apical localization of APOB. They therefore conclude that ANKRD9 mediates the regulation of APOB-containing chylomicron process by ATP produced through purine biosynthesis/salvage pathway.

Overall, the results are interesting and mostly convincing. However, many of the conclusions drawn by the authors lack sufficient evidence, and some results were not explained or discussed. The underlying mechanisms by which ANKRD9 regulates APOB trafficking and purine metabolism need to be further explored.

Major comments:

1. Fig. 3 compared subcellular localization of APOB in WT and Ankrd9 KO jejunum. This is an important piece of data. The authors need to provide evidence showing that the indicated antibody (ab7616) can indeed specifically label mouse APOB48/100 by performing WB using jejunal lysates. Why is APOB staining in KO mice in Fig. 3A different from that in Fig. 3C? Is it because mice in 3A were not subjected to oil gavage? If that were the case, the legends (as well as the Methods) need to be expanded to include how mice were treated (fasted or fed oil)? A similar validation of anti-APOB antibody for APOB IHC staining in Caco2 cells is also required.
2. Fig. 4B and S5D: ANKRD9 apparently stains the apical and lateral surfaces as well. Can the authors comment on this? If not NTD, which domain of ANKRD9 mediates its association with the membrane?
3. The mechanisms how membrane-associated ANKRD9 promotes APOB trafficking should be further explored. The images showing GM130 IHC staining in WT and Ankrd9 KO jejunum need to be included.
4. Reduced IMP levels can explain for reduced levels of AMP and GMP and eventually ATP and GTP in Ankrd9 KO enteroids, despite that IMPDH2 becomes stabilized. How can ANKRD9 regulate IMP levels or, if not, other enzymes/intermediates in purine biosynthesis/salvage pathway? AlphaFold-based prediction of ANKRD9-PRPS1 interaction was nice but preliminary and needs to be verified by co-IP assay. Moreover, how exactly can ANKRD9-PRPS1 interaction contribute to purine biosynthesis/salvage pathway? Thru PRPS1 stabilization?
5. Is ANKRD9 expression subjected to lipid loading regulation in vivo and in vitro?
6. The reversal of APOB localization by overexpression of ANKRD9 in ATP-depleted cells does not necessarily support the conclusion in lines 361-362, unless evidence showing ATP levels are elevated by ANKRD9 overexpression is provided.
7. Fig. S13 adds little value to the study and can be removed.
8. Can the authors discuss why Ankrd9 KO mice have a lean body phenotype?

Minor comments:

1. ANKRD9 is not specifically expressed in the mouse intestine based on Fig. 1C and Biogps. Intestine-specific KO seems to be a better strategy than global KO to examine the effects of ANKRD9 on intestinal lipid metabolism. Can the authors explain why not using intestine-specific KO mice for the study?
2. Lines 81-86: These statements appear to be the summary of current study and better not to be placed in the first paragraph of Introduction.
3. Please include a scheme showing the functional domains of ANKRD9 in Fig. 1 as well.
4. Figure legends need to be expanded, at least for the mouse work. For example, were mice in Fig. 1I and J fasted and then gavaged with oil?
5. The BODIPY fluorescence intensity at 90 min and 120 min post oil feeding should also be quantified and included in Fig. 2C.

Reviewer #4

(Remarks to the Author)

(General comments)

This manuscript uncovers a novel mechanism whereby the scaffolding protein ANKRD9 regulates ATP production via the purine biosynthesis/salvage pathway, which is critical for dietary fat absorption in enterocytes. Especially, authors identified ANKRD9 as a novel link between purine metabolism and the intracellular trafficking of ApoB/chylomicrons. Furthermore, they discovered the dual role of ANKRD9, both in the cytosol (regulating IMPDH2 via its N-terminal domain) and in membrane-associated complexes near cis-Golgi, is mechanistically helps to reconcile previous conflicting data about

ANKRD9 function. To demonstrate these findings, authors used global Ankrd9 knockout mice, high-resolution imaging (confocal and EM), proteomics and metabolomics, functional rescue, and temporal tracking of ApoB localization. Overall, study is well designed and performed. However, authors should address several issues such as lack of quantification in several images to improve the quality of this manuscript, as shown below.

(Specific comments)

1. The finding that Ankrd9 deletion leads to reduced intestinal ATP levels despite preserved mitochondria and glycolysis is novel. Please include in the Discussion how purine-derived ATP compensates or complements these more traditional energy-generating pathways.
2. The effects of Ankrd9 deletion on lipid absorption are convincingly shown to result in leaner phenotypes, but long-term metabolic outcomes (e.g., response to high-fat diet) are not explored.
3. Figure 1H–J (TG quantification and lipid staining): Oil Red O and BODIPY images are appropriately scaled, but quantification of lipid droplet number or area per enterocyte should be included. In addition, the localization of lipids near the nucleus is an important finding. Authors should include the high-magnification insets to show this subcellular detail more clearly.
4. Figure 2C (Quantification of lipid uptake): The method used for fluorescence intensity quantification and how representative images were selected should be described in the legend or methods to avoid bias.
5. Figures 3A & 3C (ApoB localization): Quantitative data on ApoB localization (e.g., percent colocalization with cis-Golgi, apical vs lateral membrane ratios) should be included, as a panel D.
6. Figure 4C-D (Native PAGE for protein complexes): Densitometric quantification of band intensities should be included to assess changes in complex abundance and data repeatability.
7. Figure 5C-H (ATP/GTP quantification): Please clarify whether nucleotide levels are normalized per cell, per protein, or per volume in the y-axis label.
8. Figure 6: Direct mechanistic insight into how ANKRD9 modulates PRPS1 or other purine enzymes (besides IMPDH2) is based on AlphaFold predictions, not biochemical validation. Thus, co-IP or proximity labeling experiments should be performed to strengthen the mechanistic insights.
9. Figure 7 (ANKRD9 localization): Quantification of puncta number or intensity over time as well as Golgi marker should be included in this figure to confirm localization shift relative to Golgi compartments.
10. Figure 8 (IMPDH2 and rescue experiments): Rescue experiments using adenosine supplementation need a quantification and statistical analysis.

Version 1:

Reviewer comments:

Reviewer #1

(Remarks to the Author)

I am very happy that the authors were receptive and responsive to my comments. I am satisfied with their responses, the revised manuscript as well as the supplemental material. I recommend this revised manuscript be accepted for publication.

Reviewer #3

(Remarks to the Author)

The authors have addressed my concerns satisfactorily.

Reviewer #4

(Remarks to the Author)

Authors responded to this reviewer's previous concerns satisfactory by performing additional experiments and data analysis. No further comments.

Response to Reviewer 1

Critique: This is potentially an interesting topic which is not very well understood. Despite my enthusiasm for the topic, I have many problems with the paper which greatly diminished my enthusiasm for the paper.

We thank the reviewers for his/her interest in the topic, for critical reading of the manuscript and comments that helped us to clarify and improve this work. In the revised manuscript, we addressed the reviewer's concerns, as detailed below.

1. This is a very complex paper to read and to understand. I have read the manuscript carefully a couple of times, but I still have difficulty being convinced of the conclusions made by the authors.

The authors interchangeably use the term of fat absorption and nutrient absorption. They are two very different terms referring to two totally different functions carried out by the gastrointestinal tract. There are three major macronutrients in the nutrients processed by the small intestine, lipid, protein and carbohydrate. Thus, it is imperative that this is clearly defined in the manuscript. Within the process of intestinal fat absorption, there are also three major steps, uptake, resynthesis of triglyceride, and the formation and secretion of chylomicrons. Thus this paper has to be totally rewritten before it should be reviewed by any journal.

We do appreciate the reviewer's point about the distinct mechanisms of absorption for various nutrients. Consequently, we carefully edited the entire manuscript to focus specifically on the dietary absorption of fat.

2. When the authors are discussing intestinal fat absorption, they should clearly define which component of intestinal fat absorption they are talking about. Some concepts in the paper are not accurate. For instance, the authors referred to the uptake of fatty acids by the small intestinal epithelial cells as being active. This is not correct. Most investigators in the field of fat absorption would consider the initial uptake of fatty acids as being passive and is driven by the concentration gradient of fatty acids between the lumen and the intracellular compartment.

The reviewer is correct that the fatty acid uptake is driven by concentration, and we edited our text for clarity.

3. Histology (immunohistochemistry) and electron microscopy are major techniques utilized in this study. While these are accepted tools in many biological and physiological research, an important consideration is the possibility of misinterpretation, because of the small number of samples studied. For instance, how many sections are studied in each animal and the statistical analysis are obviously important considerations. Just studying 3 animals per group is not rigorous enough for one to draw conclusions about an observation. This important question is not addressed at all in the manuscript. This is a major weakness of the manuscript.

We regret not including these details in the original manuscript. In the EM experiments, for each animal, 4-5 pieces were cut out of jejunum tissue to prepare grids. Then, 2-3 sections per grid were analyzed, and a total of 25-35 images were collected per mouse. This allowed us to quantify Golgi size from 80-90 cells per group. These details have been added to the Methods section. The revised figure legends state that for each histology and immunohistochemistry experiment, data were collected from at least 4-5 mice per genotype and 8-9 sections were analyzed for each animal. Additionally, we quantified the fluorescence intensity images and

verified the immunohistochemistry and histology results by direct measurements of lipids using mass spectrometry. These new data are included in the revised manuscript as Fig. 2c-d.

4. In figure 2, the investigators harvest the jejunal tissue for histology at different time points after the oral administration of olive oil. As pointed out above in 3, I am not sure if the fluorescence intensity data is totally accurate without knowing how many tissue sections were examined in each animal. How reproducible is the observation from the wild type and the knockout animals.

As described in the figure legend, in these experiments, we used at least 3 mice per each time point per each group. The experiment was repeated *independently* 3 times. Therefore, the images shown in Fig. 2b and 3c are representative data of at least 9 sections per each time point in each wild-type and knockout categories.

Also, it is a real pity that the investigator did not harvest any plasma samples in the animals to measure triglyceride level. This would provide them with a good measure of the amount of lipids transported by the gut at different time points after oleic acid feeding.

We collected the intestinal tissue and serum during the oil gavage experiment and analyzed them by mass-spectrometry. We appreciate the reviewer's suggestion to include these data. Since we used oleic acid (OA) in the feeding experiment, and OA was, by far, the most abundant lipid, we calculated changes in oleic acid 18:1 and oleic acid-containing triglycerides in the intestine and the serum. The results revealed the same rate of oleic acid uptake, but an initial delay in triglyceride synthesis in the KO intestine. This result supports our original conclusions based on imaging studies that the processing of dietary fat through the secretory pathway is initially delayed in the KO enterocytes due to low ATP content. With time, the levels of triglycerides in the KO and WT become similar. These data are shown in (Fig. 2d and Supplementary Fig. 4b-c).

Furthermore, most investigators studying intestinal lipid absorption and transport monitor the plasma lipid levels for 4 hours. How can the investigators be sure that 1 hour was the ideal length of the study. The fluorescence was still going up between 30 – 60 mins in the knockout animals while the WT animals was decreasing between 30 – 60 mins. This is a major deficiency in experimental design.

We respectfully disagree. An increase in ATP utilization occurs as soon as fatty acids enter the cell, and our goal was to understand which of the intracellular lipid processing steps was significantly affected by the loss of Ankrd9. The 0, 15 min, 30 min, and 60 min time points were chosen because, in kinetic studies, early time points provide a more precise measure of rates, as the measured values reflect the fatty acid uptake, triglyceride synthesis, and ApoB intracellular trafficking before the incoming nutrients became sufficiently metabolized to compensate for the energy deficits caused by the loss of Ankrd9.

5. The quality of the histology and immunohistochemical images were not particularly good. For instance, figure 3 of the apoB48 staining was not particularly convincing. I am not convinced that one can conclude with confidence what the images are telling us about a particular physiological process.

It is not clear which aspects of ApoB staining the reviewer questions. Our images of ApoB in control intestine do not seem to differ significantly from the published data (see PMID: 21805327, PMID: 12493769, PMID: 28958857, and PMID: 14565984). To better illustrate

changes in the ApoB pattern, we used a plot profile tool in ImageJ, quantified and compared the ApoB signal intensity across enterocytes at different time points (This quantitation is now included as Fig. 3d). To increase the reviewer' and readers' confidence in the specificity of staining, we now include Western blots of mouse intestinal tissue and enteroids, which show a single band of expected size (new Supplementary Fig. 4e-f). We also include Western blot of non-differentiated and differentiated Caco-2 cells to illustrate the expected upregulation of ApoB (new Supplementary Fig. 5a).

6. The apo B 48 data should be interpreted with caution. Yes, apo B with associated with chylomicrons and there is one apo B per chylomicron particle. It has been demonstrated elegantly by Davidson and Glickman from Columbia University apo B secretion is not altered by the amount of lipid secreted in chylomicrons. This observation has been confirmed by other investigators using the lymph fistula rat model. Thus, the movement of apo B should not be interpreted as chylomicron formation or secretion. The only way to directly quantitate the amount of chylomicron triglyceride secreted is by the lymph fistula mouse model. Though technically difficult, it is certainly doable.

We do not claim that ANKRD9 deletion alters ApoB secretion into the blood. Instead, we show that the intracellular trafficking of ApoB (which is the major component of chylomicrons) is altered in *Ankrd9*^{-/-} enterocytes due to low levels of purine nucleotides. The main conclusion of our study is that ANKRD9-dependent purine biosynthesis/salvage provides energy necessary for efficient intracellular trafficking of lipid and ApoB. We have edited the manuscript to make this conclusion clearer.

Response to Reviewer 3

The study by Wang and colleagues identified ANKRD9 as an important regulator of dietary lipid metabolism in the mouse intestine. They started with the observations that *Ankrd9* KO jejunum had robust lipid accumulation compared with the controls. They demonstrated that ANKRD9 was concentrated near the cis-Golgi and promoted APOB transport through the Golgi in response to lipid uptake. Interestingly, they found that ANKRD9 also regulated nucleotide metabolism, and that *Ankrd9* KO jejunum/enteroids had lower levels of ATP, GTP and IMP. The authors further showed that lipid loading increased intracellular ATP levels in an ANKRD9-dependent manner, and that intracellular ATP levels conferred apical localization of APOB. They therefore conclude that ANKRD9 mediates the regulation of APOB-containing chylomicron process by ATP produced through purine biosynthesis/salvage pathway.

Overall, the results are interesting and mostly convincing. However, many of the conclusions drawn by the authors lack sufficient evidence, and some results were not explained or discussed. The underlying mechanisms by which ANKRD9 regulates APOB trafficking and purine metabolism need to be further explored.

We thank the reviewer for his/her interest in our work, the careful reading of the manuscript, and constructive suggestions that help to increase clarity and mechanistic insight. Below, please find the point-by-point response to the critique.

1. Fig. 3 compared the subcellular localization of APOB in WT and *Ankrd9* KO jejunum. This is an important piece of data. The authors need to provide evidence showing that the indicated antibody (ab7616) can indeed specifically label mouse APOB48/100 by performing WB using

jejunal lysates... A similar validation of anti-APOB antibody for APOB IHC staining in Caco2 cells is also required.

This is an important point. In the revised manuscript, we include the Western blot analysis of jejunum and jejunal enteroids using anti-Apob antibody (ab7616, Abcam) As shown in the new Supplementary Fig. 4e-f and 5a, the antibody recognizes a single band of appropriate size (~250kDa). Using the same antibody we also show that ApoB protein was not detected in non-differentiated Caco2 cells that do not express ApoB, but detects the right band in differentiated cells, further confirming specificity (new Supplementary Fig. 5a).

...Why is APOB staining in KO mice in Fig. 3A different from that in Fig. 3C? Is it because mice in 3A were not subjected to oil gavage? If that were the case, the legends (as well as the Methods) need to be expanded to include how mice were treated (fasted or fed oil)?

The reviewer is correct. Data shown in Fig. 3a were collected for animals under non-fasting conditions, whereas those shown in Fig. 3c show results after 6h fasting. We edited the figure legends and methods sections to include this information.

2. Fig. 4B and S5D: ANKRD9 apparently stains the apical and lateral surfaces as well. Can the authors comment on this?

To better illustrate ANKRD9 localization and targeting, we performed confocal sectioning of differentiated Caco2 cells, immunostained for endogenous ANKRD9, the apical membrane marker and the Golgi marker. Secondly, we expressed the recombinant ANKRD9 in *Ankrd9*^{-/-} cells and performed similar Z-sectioning/immunostaining experiments. In both conditions, we observed strong enrichment of ANKRD9 near the lateral membrane and near Golgi. The ANKRD9 puncta can also be found between the Golgi and apical membrane; however, colocalization with the apical membrane was not detected. These data are included in the revised manuscript as Fig. 4b, 5c-d and Supplementary Fig. 6a-b.

Part 2: If not NTD, which domain of ANKRD9 mediates its association with the membrane?

Thank you for this question, which helped to clarify the basis of ANKRD9 targeting to the membrane. ANKRD9 forms large proteinaceous particles, and since the size of the complex does not change in response to N-terminal deletion, the density-based centrifugation may not be sensitive enough to determine whether the complex remains targeted or simply sediments based on size and density. As an alternative approach, we used jejunal enteroids from *Ankrd9*^{-/-} mice (to avoid potential interference with the endogenous *Ankrd9*) and expressed the full-length mouse *Ankrd9*, human full-length ANKRD9, and human ANKRD9 (1-63 del). The full-length ANKRD9/*Ankrd9* localized, as expected, in puncta enriched near the Golgi and the lateral membrane. The ANKRD9 (1-63 del) variant formed puncta in the cytosol with little or no lateral membrane localization. These results are included in Fig. 4b and Supplementary Fig. 6a.

3. The mechanisms how membrane-associated ANKRD9 promotes APOB trafficking should be further explored. The images showing GM130 IHC staining in WT and *Ankrd9* KO jejunum need to be included.

The anti-GM130 antibody (BD610823) does not work in the intestinal tissue, consequently to address the reviewer's recommendation, we conducted additional experiments testing how loss of *Ankrd9* affects Golgi morphology at steady state and in response to fat. Using confocal Z-sectioning we found that control enteroids (in addition to a centrally located Golgi network)

contain numerous GM130 positive vesicles near the apical membrane; these vesicles are missing in *Ankrd9*^{-/-} enteroids and reappear in Ankrd9 cells transfected with the full-length ANKRD9 but not ANKRD9 N-terminal deletion (new Fig. 5c-d and Supplementary Fig. 6b). These results, together with the “swollen” of cis-Golgi that we initially observed by EM in *Ankrd9*^{-/-} enterocytes (Fig. 5a) and a different timing of Golgi network expansion in response to oleic acid (Fig. 5e) suggest that ANKRD9 directly or indirectly (by changing local nucleotide levels) regulates Golgi dynamics.

4. Reduced IMP levels can explain for reduced levels of AMP and GMP and eventually ATP and GTP in Ankrd9 KO enteroids, despite that IMPDH2 becomes stabilized. How can ANKRD9 regulate IMP levels or, if not, other enzymes/intermediates in purine biosynthesis/salvage pathway? AlphaFold-based prediction of ANKRD9-PRPS1 interaction was nice but preliminary and needs to be verified by co-IP assay. Moreover, how exactly can ANKRD9-PRPS1 interaction contribute to purine biosynthesis/salvage pathway? Thru PRPS1 stabilization?

To answer this question comprehensively and without bias, we would need to measure metabolic fluxes. Currently, we are not set up for such experiments. Our attempts to identify ANKRD9 interacting partners have been complicated by the need to conduct studies in differentiated polarized monolayers, where ANKRD9 has proper localization, and using low expression of recombinant ANKRD9 (to avoid changes in the nucleotide balance and preserve physiologically relevant interactions). These conditions so far have not yielded a sufficient amount of material for reliable identification of interacting proteins by mass-spectrometry.

Consequently, to provide additional insights, we tested whether (i) ANKRD9 directly interacts with PRPS1, (ii) regulates PRPS1 abundance, and/or (iii) modifies the intracellular pattern of PRPS1. We found that the deletion or mild overexpression of ANKRD9 does not change PRPS1 protein abundance. The co-IP experiments yield a specific but weak PRPS1 signal (new Supplementary Fig. 12c). However, analysis of PRPS1 puncta revealed that the number of PRPS1-positive particles larger than 0.03 μm^2 was significantly lower in the jejunum of *Ankrd9*^{-/-} mice compared to the wild-type controls (new Fig. 7e-f). Changes in PRPS1 oligomeric state and interaction with other proteins affect PRPS1 activity (PMID: 29074724); and the ability of PRPS1 to form higher-order oligomers/filaments was previously reported (PMID: 37248548). We hypothesize that ANKRD9 modulates PRPS1 activity by regulating the size of PRPS1 protein complexes.

5. Is ANKRD9 expression subjected to lipid loading regulation in vivo and in vitro?

We tested whether the 16-hour fasting (relative to feeding chow containing 10% fat) alters the ANKRD9 mRNA levels and found no statistically significant changes. So far, our data suggest that lipid-loading modulates ANKRD9 localization (and potentially protein-protein interactions) rather than overall abundance.

6. The reversal of APOB localization by overexpression of ANKRD9 in ATP-depleted cells does not necessarily support the conclusion in lines 361-362, unless evidence showing ATP levels are elevated by ANKRD9 overexpression is provided.

We have included the analysis ATP level in ANKRD9-overexpressed Caco2 cells (now in Supplementary Fig. 14b).

7. Fig. S13 adds little value to the study and can be removed.

While we agree with the reviewer, our co-authors put considerable effort into generating this figure, so we prefer to keep it (now Supplementary Fig. 15)

8. Can the authors discuss why *Ankrd9* KO mice have a lean body phenotype?

The lean body phenotype observed in *Ankrd9* knockout mice could be explained by (i) slow fat processing in the intestine, resulting in the intestinal fat retention and/or (ii) high abundance of *Ankrd9* in heart and skeletal muscle. Loss of *Ankrd9* in these tissues may cause energy deficits similar to the those in the intestine, forcing *Ankrd9*^{-/-} mice to utilize body fat as an energy source to sustain physiological processes under fasting conditions. Although deletion of *Ankrd9* does not affect random blood glucose levels (Supplementary Fig. 2c), it significantly reduces fasting (16 h) blood glucose levels (Supplementary Fig. 2d) in agreement with higher demand for metabolic fuels by *Ankrd9*^{-/-} tissues during fasting.

Minor comments:

1. ANKRD9 is not specifically expressed in the mouse intestine based on Fig. 1C and Biogps. Intestine-specific KO seems to be a better strategy than global KO to examine the effects of ANKRD9 on intestinal lipid metabolism. Can the authors explain why not using intestine-specific KO mice for the study?

Our work is the first study to report the physiological function of *Ankrd9* *in vivo*. When we started, we did not know which tissue would show a phenotype and what phenotype would look like. We agree that the next steps should be targeted deletion of ANKRD9 in specific cells/tissues.

2. Lines 81-86: These statements appear to be the summary of current study and better not to be placed in the first paragraph of the Introduction.

The revised manuscript has been reformatted using *Nat Commun* guidelines.

3. Please include a scheme showing the functional domains of ANKRD9 in Fig. 1 as well.

Thank you for the suggestion. The updated Fig. 1a, and corresponding legend highlight the known domains of ANKRD9.

4. Figure legends need to be expanded, at least for the mouse work. For example, were mice in Fig. 1I and J fasted and then gavaged with oil?

We have edited the methods and figure legends to incorporate additional details: The intestines shown in Figures 1i and 1j were harvested during the daytime under non-fasting conditions. Oil Red O and BODIPY stainings were performed on different sets of mice. For each set of experiments, data represent 4-6 animals from the wild-type and knockout groups, respectively.

5. The BODIPY fluorescence intensity at 90 min and 120 min post oil feeding should also be quantified and included in Fig. 2C.

We have included the quantification of the fold change in BODIPY fluorescence area per cell during lipid treatment (from 0 min to 120 min) in Fig. 2c.

Response to Reviewer 4

(General comments)

This manuscript uncovers a novel mechanism whereby the scaffolding protein ANKRD9 regulates ATP production via the purine biosynthesis/salvage pathway, which is critical for dietary fat absorption in enterocytes. Especially, authors identified ANKRD9 as a novel link between purine metabolism and the intracellular trafficking of ApoB/chylomicrons. Furthermore, they discovered the dual role of ANKRD9, both in the cytosol (regulating IMPDH2 via its N-terminal domain) and in membrane-associated complexes near cis-Golgi, is mechanistically helps to reconcile previous conflicting data about ANKRD9 function. To demonstrate these findings, authors used global Ankrd9 knockout mice, high-resolution imaging (confocal and EM), proteomics and metabolomics, functional rescue, and temporal tracking of ApoB localization. Overall, study is well designed and performed. However, authors should address several issues such as lack of quantification in several images to improve the quality of this manuscript, as shown below.

We thank the reviewer for his/her interest in our work and constructive suggestions that helped to improve the manuscript. Our point-by-point responses are as follows:

1. The finding that Ankrd9 deletion leads to reduced intestinal ATP levels despite preserved mitochondria and glycolysis is novel. Please include in the Discussion how purine-derived ATP compensates or complements these more traditional energy-generating pathways.

Thank you for this suggestion. We have now included such a discussion. Specifically, our data indicate that ANKRD9-dependent regulation of purine balance is important during the initial steps of lipid processing to compensate for the rapid use of ATP for triglyceride synthesis and ApoB trafficking.

2. The effects of Ankrd9 deletion on lipid absorption are convincingly shown to result in leaner phenotypes, but long-term metabolic outcomes (e.g., response to high-fat diet) are not explored.

We agree that going forward, testing various diets, especially high-fat diet, would be very interesting and informative. The time frame for the revision did not allow us to conduct such a study.

3. Figure 1H–J (TG quantification and lipid staining): Oil Red O and BODIPY images are appropriately scaled, but quantification of lipid droplet number or area per enterocyte should be included. In addition, the localization of lipids near the nucleus is an important finding. Authors should include the high-magnification insets to show this subcellular detail more clearly.

Thank you for this suggestion. We included the quantification results in Fig. 1i-l along with the zoomed-in images.

4. Figure 2C (Quantification of lipid uptake): The method used for fluorescence intensity quantification and how representative images were selected should be described in the legend or methods to avoid bias.

We agree. In the revised manuscript, we clarified the quantitation method in the Fig. 2c legend.

5. Figures 3A & 3C (ApoB localization): Quantitative data on ApoB localization (e.g., percent colocalization with cis-Golgi, apical vs lateral membrane ratios) should be included, as a panel D.

In response to this recommendation, we included the quantification results in Fig. 3d.

6. Figure 4C-D (Native PAGE for protein complexes): Densitometric quantification of band intensities should be included to assess changes in complex abundance and data repeatability.

We have now included the quantification in Fig. 4c.

7. Figure 5C-H (ATP/GTP quantification): Please clarify whether nucleotide levels are normalized per cell, per protein, or per volume in the y-axis label.

As described in the method section, the nucleotide levels are normalized per protein amount.

8. Figure 6: Direct mechanistic insight into how ANKRD9 modulates PRPS1 or other purine enzymes (besides IMPDH2) is based on AlphaFold predictions, not biochemical validation. Thus, co-IP or proximity labeling experiments should be performed to strengthen the mechanistic insights.

We agree and did additional experiments. Specifically, we tested whether (i) ANKRD9 directly interacts with PRPS1, (ii) regulates PRPS1 abundance, and/or (iii) modifies the intracellular pattern of PRPS1. We found that the deletion or mild overexpression of ANKRD9 does not change PRPS1 protein abundance. The co-IP experiments yield a specific but weak PRPS1 signal (new Supplementary Fig.12c). However, analysis of PRPS1 pattern revealed that the number of PRPS1-positive puncta larger than $0.03 \mu\text{m}^2$ was significantly lower in the jejunum of *Ankrd9*^{-/-} mice compared to the wild-type controls (new Fig. 7e-f). PRPS1 is a soluble protein, but can form higher-order oligomers/filaments (PMID: 37248548). Furthermore, changes in PRPS1 oligomeric state and interaction with other proteins affect PRPS1 activity (PMID: 29074724). We hypothesize that ANKRD9 modulates PRPS1 activity by regulating the size and/or composition of PRPS1 large protein complexes.

9. Figure 7 (ANKRD9 localization): Quantification of puncta number or intensity over time as well as Golgi marker, should be included in this figure to confirm localization shift relative to Golgi compartments.

New Fig. 8 includes quantification of time-dependent changes in Gm130/ANKRD9 colocalization during oleic acid treatment.

10. Figure 8 (IMPDH2 and rescue experiments): Rescue experiments using adenosine supplementation need quantification and statistical analysis.

We have included quantification and statistical analysis of IMPDH2 oligomerization into a new Fig. 9 (former Fig. 8).

Since the revised manuscript now includes additional rescue data using the expression of ANKRD9 and N-terminally deleted ANKRD9 in *Ankrd9*^{-/-} cells (Fig. 4b, Fig. 5c-d and Supplementary Fig. 6a-b), we deleted adenosine supplementation experiment.